# Dynamic Dual-Feedback Conformal Inference for Time series Forecasting

## Abstract

Quantifying uncertainty in time series forecasting is particularly demanding because sequential data exhibit temporal dependence and are prone to distributional changes. Conformal inference has emerged as a powerful uncertainty quantification approach for evaluating the reliability of predictive models through the construction of prediction sets. Recent advances have introduced online conformal methods that adaptively adjust prediction thresholds through feedback mechanisms. However, the existing feedback mechanism typically relies solely on miscoverage indicators (actual feedback)—whether the true label falls within the interval at each time step—while overlooking the empirical prediction threshold (estimated feedback) that is derived from the oracle conformal method. In this paper, we propose *Dynamic Dual-feedback Conformal Inference* (DDCI), which incorporates a dual-feedback mechanism consisting of *actual feedback* and *estimated feedback*. The former drives the primary adjustment of the intervals based on true observations, while the latter dampens excessive expansions or contractions by leveraging empirical thresholds from conformal inference during updates. By balancing these two signals, DDCI achieves more stable and narrower prediction intervals in sequential settings while preserving the coverage validity.

## 1 Introduction

Machine learning models are increasingly being applied in many real-world domains such as health care, energy and transportation, where uncertainty quantification is essential for decision-making (Díaz-González et al., 2012; Badue et al., 2021). Generally speaking, prediction sets/intervals are a common approach to reflect models' uncertainty. However, traditional machine learning or statistical models themselves are difficult to quantify uncertainty, or require strong assumptions on the data distributions (Durbin & Koopman, 2012; Gasthaus et al., 2019; Salinas et al., 2020).

Conformal inference, firstly introduced by Vovk et al. (1999), is a non-parametric framework to construct prediction sets, which enjoys statistically marginal coverage guarantee under a lenient assumption called exchangeability. This framework offers a general methodology for converting the outputs of black-box models into prediction sets and can be applied on top of any underlying predictive model (Angelopoulos et al., 2023b). The simplicity and generality of conformal methods have led to their wide adoption in real-world applications (Eklund et al., 2015; Luo et al., 2024). Fundamentally, the prediction sets generated by conformal methods are guaranteed to contain the true response with a user-defined probability. However, exchangeability often fails to hold in sequential data (e.g., time series) and the validity of conformal inference may be compromised because of distributional changes arising from external factors such as seasonality or external events. Motivated by this challenge, this work studies the problem of conformal inference for time series data.

In recent years, substantial effort has been devoted to developing conformal methods tailored for non-exchangeable time series data to achieve the target coverage guarantee. A prominent line of research is to update the prediction interval via feedback mechanisms. The very first method, Adaptive Conformal Inference (ACI), attains the desired coverage rate by dynamically adjusting target quantiles, treating distributional shifts as a single-parameter learning problem (Gibbs & Candes, 2021). Following that, multiple methods have been designed to improve the predictive efficiency (Zaffran et al., 2022; Bhatnagar et al., 2023; Angelopoulos et al., 2023a). These methods update the length of the interval by determining whether the true label falls within the interval (binary feedback

from current thresholds). Recently, Wu et al. (2025) proposed Error-quantified Conformal Inference (ECI), which incorporates the degree of under- or over-coverage to refine interval lengths. However, their approach does not fully exploit the magnitude of the error, thereby weakening the corrective effect of the feedback. More critically, a major limitation of existing methods is that all of them rely exclusively on the signal from the current thresholds, which makes the update process unstable and prone to fluctuations. In other words, the interval will gradually shrink until it can no longer cover the true value at a certain point in time, after which it will expand.

Due to the limitations mentioned above, we first propose a dynamic feedback function that more precisely quantifies the *actual feedback*. Secondly, to mitigate potential overreaction to this signal, we introduce an additional component, termed *estimated feedback*, which is from empirical thresholds derived from split conformal methods. This second feedback introduces a counter-signal with opposite sign to the actual feedback through the empirical thresholds, serving as a control mechanism that suppresses excessive adjustments and ensures smoother, more stable updates. Our main contributions are summarised as follows:

- We propose a novel online conformal methods *Dynamic Dual-feedback Conformal Inference* (DDCI) for time series forecasting, which is able to quantify the actual feedback precisely and is the first work to explore how empirical thresholds from conformal methods (estimated feedback) influence the updating process. Theoretically, our feedback mechanism does not require extra assumptions except an upper bound for the learning rate.

- Furthermore, we investigate the impact of different degrees of estimated feedback on the smoothness and efficiency of interval outputs. We also compare different kinds of estimated feedback from the original conformal method and the nonexchangeable conformal method and analyse the differences among them.

- We conduct considerable experiments on 5 datasets to demonstrate that, in the vast majority of cases, our proposed methods can substantially shorten the prediction intervals (improving efficiency) without affecting the target coverage rate, compared to all existing methods.

## 2 RELATED WORK

### 2.1 CONFORMAL PREDICTION

The task of uncertainty quantification for unobserved data has been widely studied (Abdar et al., 2021; Smith, 2024). Conformal inference, pioneered by Vovk et al. (2005), is a significant branch of uncertainty quantification methods. The power of conformal methods stems largely from its flexibility, simplicity and theoretical guarantee. Conformal methods are able to quantify distribution-free uncertainty under a lenient assumption of data distribution called exchangeability. At a high level, users define a nonconformity score function and compute scores on a calibration set. Prediction sets are then formed by including all observations whose true labels fall within the empirical quantiles of the calibration distribution (Papadopoulos et al., 2002). Multiple variants of conformal methods and score functions have been developed to improve the adaptivity and efficiency (Vovk, 2015; Lei et al., 2018; Romano et al., 2019; Angelopoulos et al., 2020; Barber et al., 2021) and have been applied to many areas (Stanton et al., 2023; Cresswell et al., 2024; Lekeufack et al., 2024; Zhang et al., 2024). A detailed introduction of conformal inference is provided by Angelopoulos et al. (2023b) and the theory behind it is given in Angelopoulos et al. (2024b).

### 2.2 CONFORMAL INFERENCE UNDER NONEXCHANGEABILITY

Conformal methods highly rely on the exchangeability, but it is often violated when they are applied to practical problems (e.g., sequential data). Recent works of developing conformal methods beyond exchangeability can be summarised into two categories. The first is to treat the past conformal scores non-equally by assigning weights, giving greater emphasis to those more similar to the present (Tibshirani et al., 2019; Xu & Xie, 2021; Barber et al., 2023). Recent approaches model temporal dependence and rely on asymptotic guarantees that require assumptions on the underlying data-generating process Xu & Xie (2023); Auer et al. (2023). The second is to adaptively update the significance level $\alpha$ or the interval directly based on the miscoverage feedback (Gibbs & Candes, 2021; Zaffran et al., 2022; Gibbs & Candès, 2024; Bhatnagar et al., 2023). The following

work (Angelopoulos et al., 2023a) advanced this line of research by framing PID control theory to prospectively model conformal scores in a predictive and adaptive manner for online scenarios. (Angelopoulos et al., 2024a) decays the learning rate based on time steps. Following that, the latest work (Wu et al., 2025) introduced an error-quantified term to update the interval by using a smoothed quantile loss. Although our work is highly inspired by the error-quantified term of Wu et al. (2025), it differs from two aspects: (i) our feedback mechanism is residual-aware which makes it more general to different datasets and underlying algorithms; (ii) ours also considers an estimated feedback that is from conformal inference itself, where all existing methods did not take this empirical threshold into account. Overall, our method is therefore able to produce substantially tighter prediction sets while maintaining its validity.

## 3 PROBLEM SETUP

**Conformal Inference for Time Series.** Under the background of time series forecasting, we observe a sequence of covariates $x_t \in \mathcal{X}$ and responses $y_t \in \mathcal{Y}$ for $t \in \mathbb{N} = \{1, 2, 3, ...\}$. Then, we train an underlying model $f_t(\cdot)$ on the data before time $t$. As in oracle conformal methods, the conformal score function $s_t(\cdot, \cdot)$ can be designed in a flexible manner, which is able to measure the accuracy of model's prediction at time $t$. A typical choice of score function for regression problems is using the absolute error $s_t(x, y) = |y_t - f_t(x)|$. The main goal is to construct a prediction interval:

$$C_t(x_t) = \{y_t \in \mathcal{Y} : s_t(x_t, y_t) \leq q_t^*\}, \tag{1}$$

where $q_t^*$, the empirical threshold, is the $(1 - \alpha)$-th quantile from the distribution of scores $S_{t-w \leq i < t} \in \{s_i(x_i, y_i)\}$. In principle, if the data is exchangeable, the interval output is expected to be marginally valid:

$$P(y_t \in C_t(x_t)) \geq 1 - \alpha. \tag{2}$$

Nevertheless, the challenge for time series forecasting is that the sequence data can be nonexchangeable due to external factors (e.g., distribution shifts). Therefore, a general goal is to achieve a long-run coverage guarantee that,

$$\lim_{T \to \infty} \frac{1}{T} \sum_{t=1}^{T} err_t = \alpha, \tag{3}$$

where $err_t = \mathbb{1}\{y_t \notin C_t(x_t)\}$.

**Online Conformal Inference.** The online update of prediction interval started from ACI (Gibbs & Candes, 2021) and evolved into the quantile tracking component inside of the conformal PID control method (Angelopoulos et al., 2023a), called online gradient descent (OGD). It considers an optimisation problem to minimise the quantile loss $q_t$,

$$q_{t+1} := q_t - \eta_t \nabla \ell_t(q_t) = q_t + \eta_t (err_t - \alpha), \tag{4}$$

where $\eta_t$ is a learning rate and $\ell_t(q_t) = (s_t - q_t)(\mathbf{1}\{s_t > q_t\} - \alpha)$ means the $(1 - \alpha)$-th quantile loss. Simply put, if the interval $C_t(x_t)$ fails to cover the true label at time $t$, the interval is expanded to make it conservative in the next step; otherwise, it is shortened.

## 4 PROPOSED METHOD

### 4.1 DYNAMIC DUAL-FEEDBACK CONFORMAL INFERENCE

In standard OGD update rule 4, the threshold adjustment relies solely on the binary coverage feedback ($err_t - \alpha$). While effective, this signal is too coarse as it only indicates whether the prediction interval covers the true label, without reflecting the magnitude or reliability of the error. Additionally, relying on a single feedback can make the update process unstable and inclined to volatility.

To address this limitation, we augment the update rule with two complementary forms of feedback. To clarify, $q_t$ and $q_t^*$ denote the current threshold and the estimated threshold from conformal inference respectively. The first is *actual feedback*, expressed as

$$\frac{|e_t|}{\max_{t-w \le i,j \le t} |s_i - s_j|} h(c * e_t), \qquad (5)$$

which incorporates the observed error $e_t = s_t - q_t$ and uses a squashing function $h(\cdot)$ to weight the update by the severity of miscoverage. This squashing function is capable of mapping the error to $[-1, 1]$ and is origin-symmetric, e.g., $tanh(x)$. This ensures that large deviations from the threshold lead to stronger corrective actions. In the function $tanh(cx)$, the parameter $c$ governs the steepness of the curve and the rate at which it saturates.

Furthermore, we employ a second feedback, termed *estimated feedback*, given by

$$\left(1 - \frac{|e_t^*|}{\max_{t-w \le i,j \le t} |s_i - s_j|}\right) h(c * e_t^*), \qquad (6)$$

which provides a stabilising signal based on the estimated error $e_t^* = s_t - q_t^*$ and $q_t^*$ is the empirical $(1-\alpha)$-th quantile of scores $S_{t-w \le i < t} \in \{s_i(x_i, , y_i)\}$ from split conformal method. For simplicity, we set $\max_{t-w \le i,j \le t} |s_i - s_j| = 2B$ based on the Assumption 1 (given below). This term acts as a safeguard to smooth the update since it is designed to always act in the opposite direction of the actual feedback: when the actual feedback pushes the interval to shrink, the estimated feedback counterbalances it by expanding the interval. A visualization of these two feedback is given in Fig. 1 with $tanh(x)$ as an illustrative example.

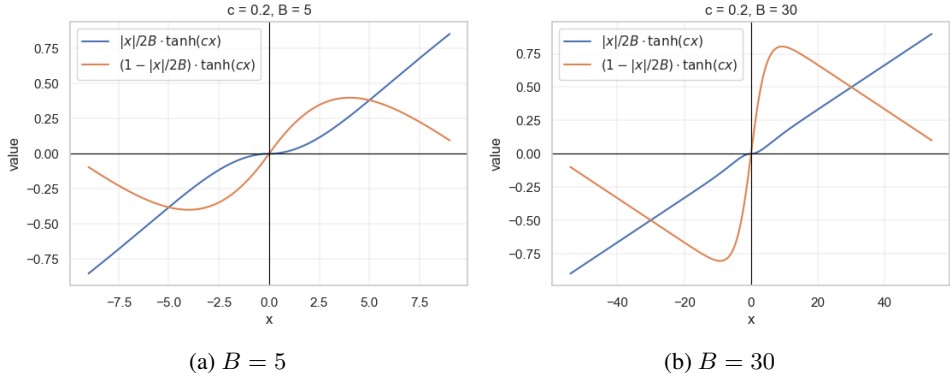

(a) $B = 5$            (b) $B = 30$

Figure 1: Plots of actual feedback function $\frac{|x|}{2B} \tanh(cx)$ (blue) and estimated feedback function $(1 - |x|/2B) \tanh(cx)$ (orange) for different values of $B$ when $c = 0.2$.

Therefore, we propose *Dynamic Dual-feedback Conformal Inference* (DDCI), based on the OGD method 4 and the novel dual-feedback mechanism, the update rule is given as follows:

$$q_{t+1} := q_t + \eta_t(\text{err}_t - \alpha) + \eta_t(\underbrace{\frac{|e_t|}{2B} h(c * e_t)}_{\text{Actual feedback}} \pm \epsilon \underbrace{\left(1 - \frac{|e_t^*|}{2B}\right) h(c * e_t^*)}_{\text{Estimated feedback}}), \qquad (7)$$

where $\epsilon > 0$ is a parameter to regulate the influence of the estimated feedback term and the symbol $'\pm'$ indicates that the estimated feedback always carries the opposite sign to the actual feedback. As shown in Fig. 1, the actual feedback term $e_t$ is highly sensitive to sudden shifts or anomalous values of $s_t$, which can exert a strong influence on the update. However, in such case, since $q_t^*$ corresponds to the $(1 - \alpha)$-th quantile of the score distribution, the estimated feedback $e_t^*$ remains relatively small, thereby mitigating excessive interval expansion and stabilising the update process. By balancing these signals, the update adapts flexibly to distributional changes while maintaining robustness, thereby constructing prediction intervals that are both responsive and stable.

### 4.2 Long-run coverage guarantees

In this section, the theoretical analysis of the proposed method is presented. More details and proofs are given in Appendix A. At first, we start from Assumption 1, which sets a boundary for scores and is a general assumption for conformal methods (Angelopoulos et al., 2023a).

**Assumption 1** *For any $t \in \mathbb{N}_+$, there exists $0 < B < \infty$ such that $s_t \in [-B, B]$.*

Based on this assumption, there are boundaries for two types of feedback: $|s_t - q_t|$ and $|s_t - q_t^*|$, which are given in the following propositions.

**Proposition 1** *Fix an initial threshold $q_1 \in [0, B]$. Assume that $\eta_t \leq \frac{2B}{2-\alpha-\epsilon}$. The update rule 10 with arbitrary nonnegative learning rate $\eta_t$ satisfies that,*

$$-(1 + \alpha - \epsilon)M_{t-1} \leq q_t \leq B + (2 - \alpha - \epsilon)M_{t-1} \quad \forall t \geq 1, \tag{8}$$

*where $M_0 = 0$, $M_t = \max_{1 \leq r \leq t} \eta_r$ for $t \geq 1$.*

**Proposition 2** *Under Assumption 1 and Proposition 1 and $\eta_t \leq \frac{B}{2-\alpha-\epsilon}$, , we have bounds for two feedback:*

$$|s_t - q_t| \leq B + (2 - \alpha - \epsilon)M_{t-1}, \qquad \text{for any } t \geq 1,$$
$$|s_t - q_t^*| \leq 2B, \qquad \text{for any } t \geq 1.$$

Given what listed above, the following theorem shows the upper bound of *long-run average miscoverage rate* with proportional learning rates.

**Theorem 1** *Let $\{\eta_t\}_{t \geq 1}$ be an arbitrary positive sequence. Under Assumption 1 and $\eta_t \leq \frac{2B}{2-\alpha-\epsilon}$, the prediction interval generated by Equation 10 with proportional learning rate $\eta_t$ satisfies:*

$$\left| \frac{1}{T} \sum_{t=1}^{T} (err_t - \alpha) \right| \leq \frac{(B + M_{T-1}) \|\Delta_{1:T}\|_1}{T} + \frac{(1 - 2\epsilon)B + (2 - \alpha - \epsilon)M_{T-1}}{2B}, \tag{9}$$

*where $\|\Delta_{1:T}\|_1 = |\eta_1^{-1}| + \sum_{t=2}^{T} |\eta_t^{-1} - \eta_{t-1}^{-1}|$, $M_T = \max_{1 \leq r \leq T} \eta_r$.*

Remarkably, our proposed method has an upper limit for the proportional learning rate $\eta_t = \eta * \max_{t-w \leq i,j \leq t} |s_i - s_j|$. Therefore, the upper boundary for $\eta$ should be $\frac{1}{2-\alpha-\epsilon}$. Since $\alpha$ and $\epsilon$ are typically small, the learning rate $\eta$ should not exceed 0.5, which is also consistent with our experimental results. In general, such learning rate is enough for all datasets without influencing the validity of prediction intervals.

## 5 EXPERIMENTS

In this section, we first describe the datasets employed to evaluate our proposed method, the underlying algorithms, and the online conformal baselines. We then present the corresponding experimental results. For fairness of comparison, since certain baselines rely on pre-specified parameters, our methods are evaluated under the same parameter settings.

### 5.1 SETTINGS

**Datasets.** Five public datasets are used to evaluate methods: stock daily opening price (Google, Amazon, Microsoft), temperature (Delhi) and electricity demand (New South Wales). A detailed description of these datasets is provided in the Appendix B.1. Due to the limit of pages, the results for electricity dataset and additional experimental results about learning rate are given in the Appendix B.2.

**Underlying Algorithms.** Four well-known algorithms are used: Prophet, AutoRegressive (AR), Theta and Transformer. The chosen models span different paradigms of time series forecasting, making them suitable for comparative evaluation. AR and Theta are rooted in classical statistical approaches, offering transparent and well-understood baselines (Box et al., 2015; Assimakopoulos & Nikolopoulos, 2000). Prophet extends statistical decomposition with Bayesian techniques, providing flexibility to handle seasonality and structural breaks (Taylor & Letham, 2018). Finally, the Transformer exemplifies state-of-the-art deep learning, capturing long-range and nonlinear dependencies through attention mechanisms (Vaswani et al., 2017).

**Online Conformal Baselines.** (1) ACI (Gibbs & Candes, 2021): updates the intervals via adjusting the quantile threshold to select from calibration set; (2) OGD family (Bhatnagar et al., 2023): uses an iterative optimization algorithm that updates model parameters incrementally; (3) Conformal PID

Control (Angelopoulos et al., 2023a): uses control theory and is capable to model scores to adjust intervals; (4) ECI family (Wu et al., 2025): uses a smoothed quantile loss to update intervals.

**Common Setup.** We set the target miscoverage rate ($\alpha$) to 10% and construct asymmetric prediction intervals (Romano et al., 2019). For all baselines, the parameter settings are consistent with those described in the original literature and corresponding open-source implementations (Angelopoulos et al., 2023a; Wu et al., 2025). The analysis of varying significance levels is given in B.2.2.

**DDCI Setup.** The squashing function employed here is $tanh(\frac{1}{2}x)$ and the hyper-paramter $\epsilon$ is set to 0.2, providing a balance between smoothness and efficiency. For DDCI, the estimated threshold $q_t^*$ is from the original split conformal method which is under the risk of under-coverage and the size of the calibration set is equal to $T\_burnin$ which is used as a warm-up data in baselines. For DDCI-Nex, the $q_t^*$ follows the design of $NexCP$ (Barber et al., 2023), where the weights for conformal scores are $w_i = \frac{0.99^{t-i}}{\sum_{j=1}^{t} 0.99^{t-j}}, \quad 1 \le i \le t$.

**Evaluation Matrics.** (1) Average coverage rate on the test set; (2) Average width of prediction intervals on the test set; (3) The Winkler score (Winkler, 1972): it is defined as the width of the prediction interval plus an additional penalty for any observation that falls outside the interval, where the penalty grows proportionally with the magnitude of the forecast error.

## 5.2 EXPERIMENTAL RESULTS

### 5.2.1 SYNTHETIC DATASETS

To verify the performance of our method with other baselines, we constructed three synthetic datasets exhibiting different forms of non-stationarity: (1) **Random Walk Trend:** a smoothed stochastic trend with time-varying volatility, generated as $X_t = X_{t-1} + \sigma_t \eta_t, \eta_t \sim \mathcal{N}(0, 1)$, where the volatility evolves according to $\sigma_t = \sigma_0 (1 + \kappa | \sin(2\pi t/P)|)$. The time-varying term $\sigma_t$ introduces slow periodic volatility modulation, capturing smoother drift and cyclical heteroskedasticity; (2) **Exponential Trend:** a multiplicative-growth process $X_t = X_{t-1} \exp(\mu + \sigma \varepsilon_t), \varepsilon_t \sim \mathcal{N}(0, 1)$; (3) **Changepoint:** a piecewise log-linear process with two regime shifts. For changepoints $c_1 = 600$ and $c_2 = 1200$, $\log X_t = \log X_{t-1} + \mu_{(t)} + \sigma_{(t)} \varepsilon_t, \varepsilon_t \sim \mathcal{N}(0, 1)$, where $(\mu_{(t)}, \sigma_{(t)})$ vary across segments, producing moderate structural breaks.

The results are averaged over 5 runs under Prophet model and given in the Table 1. Across all three synthetic settings, DDCI consistently delivers the most efficient and stable prediction intervals while maintaining the target coverage. In Setting 1 and Setting 2 scenarios, DDCI achieves the narrowest intervals and lowest Winkler scores, with DDCI-Nex providing the second-best performance. Under the more challenging changepoint setting (Setting 3), where all methods experience increased uncertainty, DDCI still produces the most informative intervals, substantially outperforming the baselines. Overall, the results show that DDCI is robust across different forms of non-stationarity and yields clear efficiency gains over existing online conformal methods.

Table 1: Results on the three synthetic datasets generated under the Prophet model. All experiments are repeated five times, and the table reports the mean and standard deviation across runs.

| Method | Setting 1 | | | Setting 2 | | | Setting 3 | | |
|---|---|---|---|---|---|---|---|---|---|
| | Coverage (%) | Avg. width | Winkler Score | Coverage (%) | Avg. width | Winkler Score | Coverage (%) | Avg. width | Winkler Score |
| OGD | $90.0 \pm 0.12$ | $11.01 \pm 0.50$ | $12.82 \pm 0.54$ | $89.9 \pm 0.17$ | $17.28 \pm 7.59$ | $21.01 \pm 8.98$ | $85.0 \pm 4.78$ | $371.43 \pm 350.57$ | $506.07 \pm 541.8$ |
| SF-OGD | $90.7 \pm 0.16$ | $75.75 \pm 0.11$ | $84.93 \pm 0.14$ | $90.3 \pm 0.23$ | $76.83 \pm 1.50$ | $86.57 \pm 2.44$ | $89.4 \pm 0.98$ | $183.13 \pm 138.25$ | $213.06 \pm 162.6$ |
| decay-OGD | $89.7 \pm 0.82$ | $21.34 \pm 2.10$ | $23.32 \pm 2.44$ | $84.3 \pm 4.90$ | $44.05 \pm 34.24$ | $56.04 \pm 47.25$ | $59.6 \pm 22.53$ | $419.60 \pm 307.13$ | $2202.6 \pm 2148.2$ |
| PID | $89.9 \pm 0.11$ | $13.25 \pm 1.18$ | $15.19 \pm 1.33$ | $89.8 \pm 0.08$ | $18.04 \pm 6.53$ | $21.72 \pm 7.95$ | $89.8 \pm 0.13$ | $124.17 \pm 100.70$ | $144.22 \pm 115.73$ |
| ECI | $90.0 \pm 0.10$ | $13.07 \pm 1.39$ | $15.07 \pm 1.53$ | $90.0 \pm 0.12$ | $19.95 \pm 7.15$ | $30.58 \pm 11.92$ | $81.9 \pm 17.84$ | $88.85 \pm 54.96$ | $119.52 \pm 61.22$ |
| ECI-cutoff | $89.9 \pm 0.07$ | $10.64 \pm 1.06$ | $12.50 \pm 1.21$ | $90.0 \pm 0.09$ | $17.57 \pm 7.23$ | $27.63 \pm 11.52$ | $89.8 \pm 0.11$ | $123.54 \pm 100.01$ | $146.74 \pm 116.99$ |
| ECI-integral | $90.0 \pm 0.06$ | $10.90 \pm 1.06$ | $12.82 \pm 1.27$ | $90.0 \pm 0.16$ | $18.00 \pm 7.80$ | $27.87 \pm 11.89$ | $89.8 \pm 0.07$ | $125.46 \pm 101.28$ | $148.14 \pm 117.75$ |
| **DDCI** | $89.9 \pm 0.02$ | $\mathbf{8.08 \pm 0.82}$ | $\mathbf{9.82 \pm 1.04}$ | $90.0 \pm 0.19$ | $\mathbf{13.15 \pm 5.35}$ | $\mathbf{22.93 \pm 9.78}$ | $89.8 \pm 0.10$ | $\mathbf{91.36 \pm 72.59}$ | $\mathbf{111.90 \pm 84.47}$ |
| **DDCI-Nex** | $89.9 \pm 0.03$ | $\underline{8.35 \pm 0.84}$ | $\underline{10.07 \pm 1.00}$ | $89.9 \pm 0.14$ | $\underline{15.36 \pm 6.75}$ | $\underline{25.20 \pm 11.18}$ | $89.8 \pm 0.10$ | $\underline{109.52 \pm 89.26}$ | $\underline{131.36 \pm 104.59}$ |

### 5.2.2 PUBLIC DATASETS

Overall, all methods achieve coverage rates close to the target level of 90%, indicating that validity is generally well maintained across different forecasting models. However, substantial differences

are observed in terms of the average and median prediction interval widths. The results of baselines can also be found within this paper (Wu et al., 2025).

**Stock Data.** Table 2 reports the performance of all methods on the Google stock dataset at the target miscoverage rate $\alpha = 10\%$. The baseline ACI method consistently delivers coverage close to the target level but at the cost of infinite interval widths, rendering it impractical. OGD family and PID also produce relatively wide intervals, particularly under the Prophet (around 57) and Transformer models (higher than 78). Among the baselines, the ECI family performs best, producing narrower intervals, but still deteriorating significantly under the Transformer model (higher than 65).

By contrast, the proposed DDCI variants achieve the most competitive results across all settings. Under the Prophet model, DDCI and DDCI-Nex reduce the average width to around 40 without influencing the coverage rate, substantially improving upon all baselines (the lowest one is 52). For the Theta model, DDCI achieves the narrowest intervals, again surpassing the best baseline. Most notably, for the Transformer model, DDCI and its variant drastically outperform all other methods, with interval widths (around 51) almost 22% shorter than those of the best ECI variant (66).

Table 2: The experimental results on the Google stock dataset. The best result (shortest width) is marked in bold, and the second-best result is marked with an underline.

| Method | Prophet Model | | | AR Model | | | Theta Model | | | Transformer | | |
|---|---|---|---|---|---|---|---|---|---|---|---|---|
| | Coverage (%) | Avg. width | Winkler Score | Coverage (%) | Avg. width | Winkler Score | Coverage (%) | Avg. width | Winkler Score | Coverage (%) | Avg. width | Winkler Score |
| ACI | 90.0 | $\infty$ | $\infty$ | 89.8 | $\infty$ | $\infty$ | 90.5 | $\infty$ | $\infty$ | 90.2 | $\infty$ | $\infty$ |
| OGD | 89.7 | 57.60 | 74.74 | 90.7 | 33.76 | 70.96 | 89.9 | 31.49 | 47.87 | 90.1 | 109.27 | 148.76 |
| SF-OGD | 89.6 | 58.92 | 77.52 | 89.9 | 28.31 | **41.93** | 90.0 | 34.04 | 51.01 | 90.1 | 88.30 | 113.34 |
| decay-OGD | 89.9 | 77.23 | 92.36 | 90.2 | 46.53 | 58.27 | 90.2 | 55.32 | 67.89 | 89.9 | 120.25 | 133.38 |
| PID | 90.1 | 57.47 | 71.18 | 89.9 | 64.19 | 83.58 | 89.9 | 75.71 | 97.37 | 90.1 | 77.36 | 106.78 |
| ECI | 89.9 | 56.06 | 75.46 | 89.7 | 19.96 | 69.11 | 89.6 | 30.92 | 47.64 | 89.9 | 70.93 | 110.29 |
| ECI-cutoff | 89.8 | 53.12 | 67.78 | 89.7 | **19.84** | 68.98 | 89.6 | 30.71 | 47.79 | 89.9 | 66.67 | 103.35 |
| ECI-integral | 89.8 | 52.36 | 67.19 | 89.7 | 19.93 | 69.17 | 89.6 | 30.42 | 47.34 | 90.0 | 68.64 | 104.52 |
| **DDCI** | 89.8 | **39.20** | **52.76** | 89.6 | 19.92 | 68.98 | 89.4 | **29.24** | **46.38** | 90.0 | **51.05** | **85.75** |
| **DDCI-Nex** | 89.7 | 39.98 | 53.72 | 89.6 | 20.07 | 68.92 | 89.6 | 29.71 | 46.71 | 89.8 | 51.58 | 86.56 |

Fig. 2 presents a visualisation of comparison result between ECI (the state-of-the-art) and DDCI (the proposed) on the Google stock dataset using the Prophet model. Generally speaking, ECI tends to produce wider ranges, which often appears overly conservative, with noticeable fluctuations in the interval boundaries. By contrast, DDCI yields narrower and more stable intervals, reducing redundancy while maintaining strong alignment with the true series.

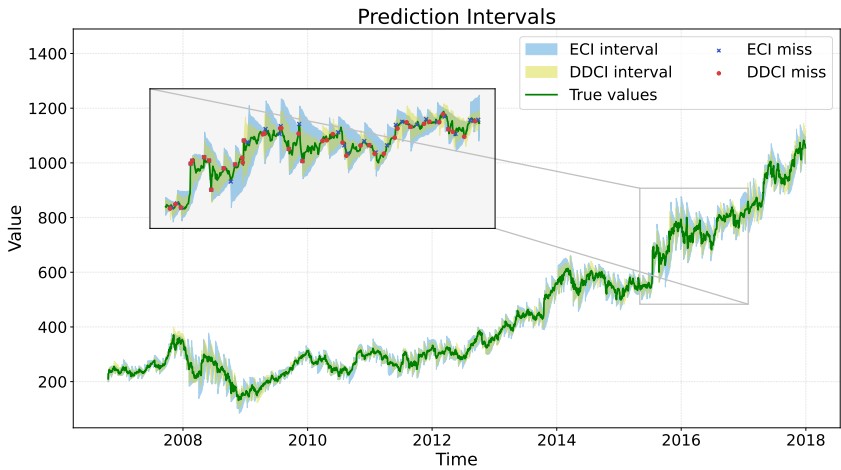

Figure 2: Visualisation of comparison results between ECI (the SOTA) and DDCI (proposed, $\epsilon = 0.2$) on Google stock dataset under Prophet model. The blue crosses and red points represent the miscovered points of each method.

In terms of Amazon data, as the table 3 presented, DDCI and its variant consistently achieve the narrowest prediction intervals while maintaining nominal coverage across all forecasting models. Improvements are particularly striking for the Prophet and Transformer models, 26% and 24% respectively. These results further confirm the superior efficiency and robustness of DDCI in generating shorter but reliable prediction intervals.

Table 3: The experimental results on the Amazon stock dataset.

| Method | Prophet Model | | | AR Model | | | Theta Model | | | Transformer | | |
|---|---|---|---|---|---|---|---|---|---|---|---|---|
| | Coverage (%) | Avg. width | Winkler Score | Coverage (%) | Avg. width | Winkler Score | Coverage (%) | Avg. width | Winkler Score | Coverage (%) | Avg. width | Winkler Score |
| ACI | 90.2 | ∞ | ∞ | 89.8 | ∞ | ∞ | 89.7 | ∞ | ∞ | 90.1 | ∞ | ∞ |
| OGD | 89.6 | 55.15 | 67.57 | 89.9 | 19.10 | **30.08** | 89.8 | 18.07 | 28.23 | 89.4 | 52.68 | 69.27 |
| SF-OGD | 89.5 | 61.47 | 73.94 | 89.9 | 24.44 | 34.31 | 90.0 | 23.88 | 34.46 | 89.3 | 56.56 | 72.48 |
| decay-OGD | 89.9 | 97.22 | 107.79 | 89.7 | 20.23 | 30.74 | 89.2 | 17.49 | 28.56 | 89.8 | 93.16 | 109.19 |
| PID | 89.8 | 45.89 | 56.02 | 89.6 | 59.22 | 76.78 | 89.5 | 71.11 | 87.88 | 89.8 | 56.06 | 71.10 |
| ECI | 90.1 | 47.00 | 59.74 | 89.5 | 17.12 | 34.23 | 89.7 | 17.46 | 27.86 | 89.9 | 49.07 | 68.77 |
| ECI-cutoff | 89.7 | 43.46 | 53.21 | 89.3 | **16.91** | 34.20 | 89.6 | 17.19 | **27.80** | 89.7 | 45.01 | 62.55 |
| ECI-integral | 89.8 | 42.01 | 51.85 | 89.5 | 16.99 | 34.25 | 89.6 | 17.20 | 27.82 | 89.7 | 45.02 | 61.35 |
| **DDCI** | 89.7 | **30.87** | 41.05 | 89.3 | 17.10 | 34.29 | 89.5 | **17.04** | 27.92 | 89.7 | 34.44 | 51.53 |
| **DDCI-Nex** | 89.7 | 31.28 | **40.96** | 89.4 | 16.99 | 34.33 | 89.6 | 17.06 | **27.80** | 89.7 | **34.10** | **51.25** |

For the Microsoft dataset (Table 4), the superiority of the proposed DDCI method over the baselines is again evident. While the ECI methods produce narrower intervals compared to earlier approaches, their widths remain consistently larger than those obtained by DDCI and DDCI-Nex across most forecasting models. In particular, DDCI variants achieve the smallest average and median widths in nearly every case, without sacrificing coverage. For example, under the Prophet and Transformer models, the intervals generated by DDCI are markedly tighter than the best baseline. These results reaffirm the consistent advantage of the DDCI method in delivering more efficient prediction intervals, further consolidating its robustness and generalisability across diverse datasets.

Table 4: The experimental results on the Microsoft stock dataset.

| Method | Prophet Model | | | AR Model | | | Theta Model | | | Transformer | | |
|---|---|---|---|---|---|---|---|---|---|---|---|---|
| | Coverage (%) | Avg. width | Winkler Score | Coverage (%) | Avg. width | Winkler Score | Coverage (%) | Avg. width | Winkler Score | Coverage (%) | Avg. width | Winkler Score |
| ACI | 90.0 | ∞ | ∞ | 90.7 | ∞ | ∞ | 89.9 | ∞ | ∞ | 89.9 | ∞ | ∞ |
| OGD | 90.0 | 3.78 | 4.80 | 90.7 | 4.37 | 4.49 | 89.9 | 2.49 | **3.31** | 89.9 | 3.80 | 5.00 |
| SF-OGD | 90.2 | 6.91 | 7.14 | 90.0 | 6.82 | 7.17 | 90.2 | 6.98 | 7.31 | 89.9 | 7.08 | 7.08 |
| decay-OGD | 90.0 | 6.02 | 5.34 | 90.1 | 4.34 | 3.64 | 90.1 | 4.91 | 4.92 | 89.7 | 5.41 | 5.41 |
| PID | 90.0 | 6.30 | 5.90 | 89.6 | 5.60 | 5.43 | 89.8 | 4.89 | 4.87 | 89.9 | 6.16 | 6.48 |
| ECI | 90.4 | 4.85 | 5.24 | 90.2 | 3.76 | 4.20 | 90.2 | 2.48 | 3.43 | 89.9 | 4.35 | 5.02 |
| ECI-cutoff | 89.8 | 4.05 | 4.59 | 89.8 | 3.04 | 3.58 | 89.8 | 2.44 | 3.39 | 89.9 | 4.08 | 4.87 |
| ECI-integral | 89.9 | 4.22 | 4.71 | 90.2 | 3.67 | 4.15 | 90.2 | 2.46 | 3.42 | 89.9 | 4.13 | 4.98 |
| **DDCI** | 89.8 | **3.26** | **3.71** | 89.8 | 2.93 | 3.58 | 89.8 | **2.39** | **3.31** | 89.9 | **3.34** | **4.10** |
| **DDCI-Nex** | 89.8 | 3.40 | 3.89 | 89.8 | **2.92** | **3.50** | 89.9 | 2.41 | 3.33 | 89.9 | 3.44 | 4.23 |

**Delhi Temperature.** On the Delhi temperature dataset (Table 5), DDCI variants consistently achieve the narrowest prediction intervals across all forecasting models while preserving nominal coverage. The improvements over the strongest baselines are smaller in magnitude compared with stock datasets but remain consistent, demonstrating the robustness and adaptability of DDCI in settings with relatively stable time series such as temperature.

In conclusion, the proposed methods generally achieve tighter prediction intervals while maintaining valid coverage. However, their improvements are less pronounced in several cases, particularly under the AR model on stock datasets and on the Climate dataset across different underlying models. These observations can be attributed to the behaviour of the conformal estimated threshold $q_t^*$. When the underlying conformal prediction already yields relatively conservative intervals, characterised by achieving the target coverage with a relatively large average width, the estimated threshold tends to overestimate local uncertainty. As a result, the deviation term $e_t^* = s_t - q_t^*$ becomes large, prompting the estimated feedback component to expand the interval more than is necessary. Such patterns are evident, for instance, in the Climate dataset, where both Prophet (coverage 90%, width 9.25) and

Table 5: The experimental results in the Delhi temperature dataset.

| Method | Prophet Model | | | AR Model | | | Theta Model | | | Transformer | | |
|---|---|---|---|---|---|---|---|---|---|---|---|---|
| | Coverage (%) | Avg. width | Winkler Score | Coverage (%) | Avg. width | Winkler Score | Coverage (%) | Avg. width | Winkler Score | Coverage (%) | Avg. width | Winkler Score |
| ACI | 91.0 | $\infty$ | $\infty$ | 90.0 | $\infty$ | $\infty$ | 90.2 | $\infty$ | $\infty$ | 90.3 | $\infty$ | $\infty$ |
| OGD | 90.4 | 7.54 | 9.78 | 90.1 | 6.82 | **9.11** | 90.0 | 6.36 | 8.68 | 89.9 | 9.72 | 13.15 |
| SF-OGD | 90.0 | 10.63 | 13.24 | 90.1 | 10.92 | 13.72 | 90.1 | 11.33 | 13.99 | 90.0 | 10.93 | 13.64 |
| decay-OGD | 90.1 | 11.24 | 13.03 | 90.0 | 10.52 | 12.25 | 89.9 | 10.95 | 12.85 | 89.7 | 14.43 | 16.18 |
| PID | 90.1 | 10.87 | 13.36 | 89.7 | 12.28 | 15.11 | 89.7 | 12.49 | 15.13 | 89.9 | 11.57 | 14.15 |
| ECI | 90.0 | 7.20 | 9.34 | 90.1 | 6.39 | 9.99 | 90.0 | 6.41 | 8.76 | 89.9 | 9.15 | 12.33 |
| ECI-cutoff | 90.1 | 7.01 | 9.24 | 90.1 | **6.29** | 9.94 | 90.0 | 6.27 | 8.72 | 89.9 | 8.77 | 12.13 |
| ECI-integral | 90.0 | 7.21 | 9.33 | 90.2 | 6.39 | 9.97 | 90.0 | 6.38 | 8.76 | 89.9 | 8.87 | 12.19 |
| **DDCI** | 90.0 | **6.95** | 9.10 | 89.9 | 6.45 | 10.04 | 90.0 | 6.36 | 8.69 | 89.9 | 8.57 | 11.80 |
| **DDCI-Nex** | 90.0 | 6.96 | **9.07** | 90.0 | 6.37 | 10.02 | 90.0 | **6.26** | **8.66** | 89.9 | **8.47** | **11.76** |

AR (coverage 89.66%, width 8.32) models produce wide yet well–calibrated intervals, and in the stock datasets under the AR model, where conservative conformal thresholds similarly restrict the gains achievable by DDCI.

### 5.2.3 SENSITIVITY ANALYSIS

We examine the role of the smoothing parameter $\epsilon$ in the update rule 10, taking Google stock data as an illustrative case. Fig. 3 shows that increasing the smoothing parameter $\epsilon$ consistently enlarges the average interval size across all four models, confirming its role in controlling the balance between efficiency and smoothness. Smaller $\epsilon$ values produce tighter intervals, while larger values lead to markedly wider intervals, with the effect most pronounced in Prophet and Transformer. Noticeably, the influence of $\epsilon$ becomes remarkable when it is greater than 0.5. However, when $\epsilon$ is small (0.2 or 0.3), although the prediction intervals become moderately wider when decreasing, this adjustment helps to reduce unnecessary misses by ensuring that more true values fall within the intervals. The apparent sensitivity shown in Figure 3 is not primarily driven by $\epsilon$ itself. The different trends across panels arise because each base model (AR, Prophet, Theta, Transformer) employs a distinct learning rate (0.05, 0.5, 0.1, 0.5, respectively), following the experimental setup of ECI. This causes the scale of the score sequence $s_t$, and consequently the update magnitude, to differ substantially across models.

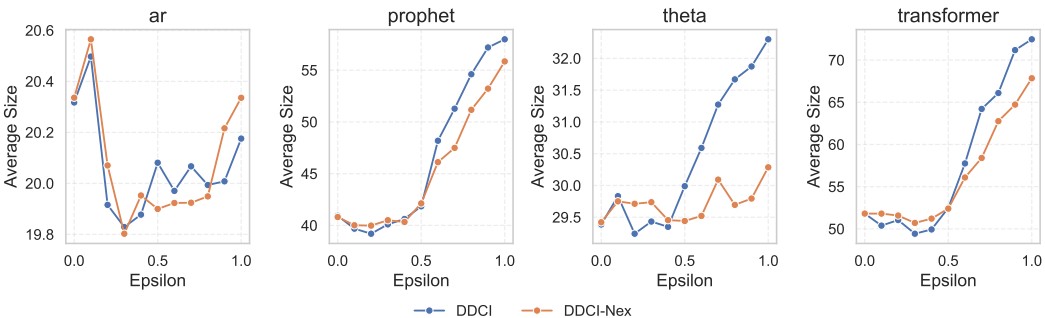

Figure 3: The influence of different smoothing parameter $\epsilon$ on the size of prediction interval.

Comparing DDCI and DDCI-Nex, DDCI is more sensitive to changes in $\epsilon$, leading to sharper interval expansion especially at higher values, whereas DDCI-Nex yields smoother and more stable interval growth. This difference arises because the estimated threshold of DDCI relies on original split conformal inference, where it is more sensitive to external factors (e.g. distribution shifts), while DDCI-Nex uses NexCP to generate the estimated threshold which provides more effective and stable adaptation. This distinction suggests that while DDCI provides stronger responsiveness to parameter tuning, DDCI-Nex offers greater robustness and stability, avoiding excessive conservativeness at large $\epsilon$. Fig. 4 illustrates the effect of different values of $\epsilon$ on the proposed DDCI method through AR model. When $\epsilon$ is small, the update process becomes more volatile, leading to fluctuations in the prediction intervals. In contrast, larger $\epsilon$ values produce smoother and more stable updates, though

at the cost of more conservative (wider) intervals. Thus, the choice of the smoothing parameter $\epsilon$ entails a trade-off between efficiency and stability.

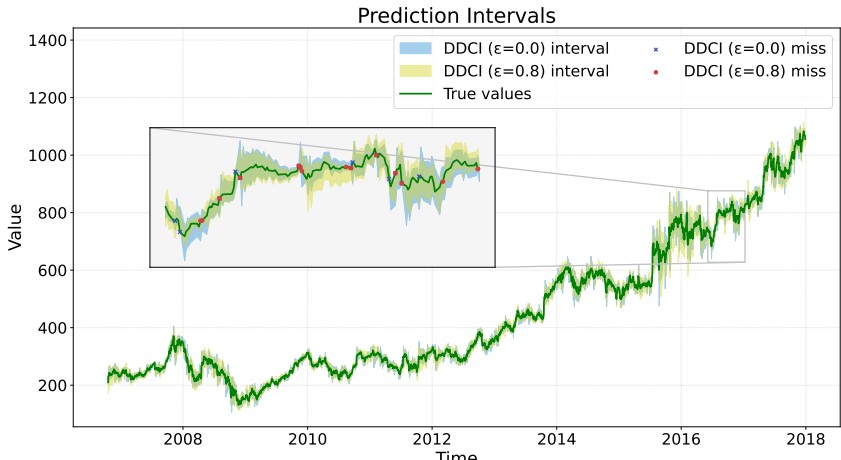

Figure 4: Visualisation of the proposed DDCI method with different $\epsilon$ (0.0 and 0.8 respectively) under AR model.

Fig. 5 displays that the effect of the smoothing parameter $\epsilon$ is relatively minor compared to its influence on interval size. Across all four models, coverage rates remain close to the target level, with only small fluctuations as $\epsilon$ increases. This indicates that varying $\epsilon$ mainly adjusts efficiency (interval width) without substantially undermining validity (coverage). Comparing the two approaches, DDCI and DDCI-Nex perform similarly in terms of maintaining coverage.

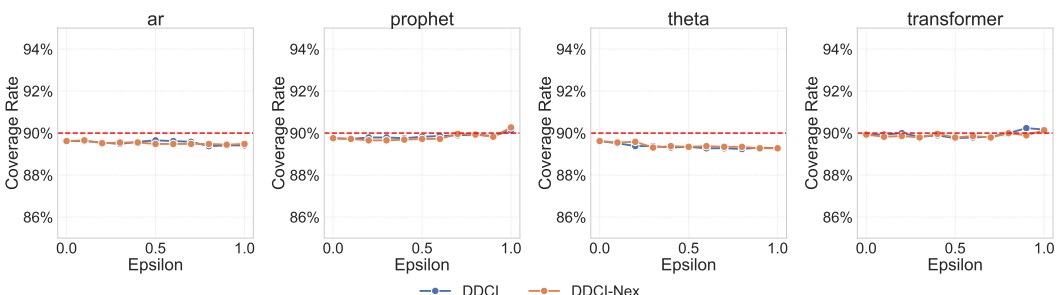

Figure 5: The influence of different smoothing parameter $\epsilon$ on the coverage rate.

## 6 CONCLUSION

Although multiple online conformal inference frameworks were proposed recently for time series forecasting, all of them are based on the *actual feedback* directly, which makes the updates fluctuated. This work introduces a novel framework that incorporates a dual-feedback mechanism. The mechanism combines *actual feedback* and *estimated feedback*, whose signs are intentionally set in opposition to counterbalance increases and decreases when updating the intervals. While the use of *estimated feedback* may slightly widen the intervals when the true label lies inside them, it effectively prevents large, unnecessary expansions, leading to more stable interval adjustments.

We test our method on three synthetic datasets and considerable public datasets under many underlying algorithms, where proves that our method can achieve tighter prediction intervals while maintaining the target coverage rate. Further work could improve the DDCI method by adaptively selecting the smoothing parameter $\epsilon$ in response to distributional changes, thereby enhancing both stability and efficiency.

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

## A    THEORETICAL ANALYSIS

Update rule:

$$q_{t+1} := q_t + \eta_t(\text{err}_t - \alpha) + \eta_t(\underbrace{\frac{|e_t|}{2B}h(c * e_t)}_{\text{Actual feedback}} \pm \underbrace{\epsilon\left(1 - \frac{|e_t^*|}{2B}\right)h(c * e_t^*)}_{\text{Estimated feedback}}), \tag{10}$$

where $\epsilon > 0$ is a parameter to regulate the influence of the estimated feedback term and the symbol $'\pm'$ indicates that the estimated feedback always carries the opposite sign to the actual feedback.

**Assumption 1** *For any* $t \in \mathbb{N}_+$*, there exists* $0 < B < \infty$ *such that* $s_t \in [-B, B]$.

Based on assumption 1, there are boundaries for two types of feedback: $e_t = |s_t - q_t|$ and $e_t^* = |s_t - q_t^*|$, which are given in the following propositions.

**Proposition 1** *Fix an initial threshold* $q_1 \in [0, B]$*. Assume that* $\eta_t \leq \frac{2B}{2-\alpha-\epsilon}$*. The update rule 10 with arbitrary nonnegative learning rate* $\eta_t$ *satisfies that,*

$$-(1 + \alpha - \epsilon)M_{t-1} \leq q_t \leq B + (2 - \alpha - \epsilon)M_{t-1} \quad \forall t \geq 1, \tag{11}$$

*where* $M_0 = 0$, $M_t = \max_{1 \leq r \leq t} \eta_r$ *for* $t \geq 1$.

*Proof.* We prove this by induction. First, $q_1 \in [0, B]$ by assumption. Assume that $\eta \leq \frac{B}{2-\alpha-\epsilon}$ and $\epsilon \geq 0$. Next fix any $t \geq 1$ and assume $q_t$ lies in the range specified in 1, and consider $q_{t+1}$.

(1): $s_t \geq q_t$,

- $q_t^* \geq s_t \geq q_t$:

$$q_{t+1} = q_t + \eta_t(1 - \alpha) + \eta_t(\frac{|s_t - q_t|}{2B}h(c * (s_t - q_t)) \pm \epsilon\left((1 - \frac{|s_t - q_t^*|}{2B})h(c * (s_t - q_t^*))\right))$$

$$\leq q_t + \eta_t(1 - \alpha) + \eta_t(1 - \epsilon)$$
$$\leq B + (2 - \alpha - \epsilon)M_t$$

- $s_t \geq q_t \geq q_t^*$:

$$q_{t+1} = q_t + \eta_t(1 - \alpha) + \eta_t(\frac{|s_t - q_t|}{2B}h(c * (s_t - q_t)) \pm \epsilon\left((1 - \frac{|s_t - q_t^*|}{2B})h(c * (s_t - q_t^*))\right))$$

$$\leq q_t + \eta_t(1 - \alpha) + \eta_t(1 - \epsilon)$$
$$\leq B + (2 - \alpha - \epsilon)M_t$$

(2): $s_t \leq q_t$,

- $q_t^* \leq s_t \leq q_t$:

$$q_{t+1} = q_t + \eta_t(\alpha) + \eta_t(\frac{|s_t - q_t|}{2B}h(c * (s_t - q_t)) \pm \epsilon\left((1 - \frac{|s_t - q_t^*|}{2B})h(c * (s_t - q_t^*))\right))$$

$$\geq q_t + \eta_t(-\alpha) - \eta_t(1 - \epsilon)$$
$$\geq -(1 + \alpha - \epsilon)M_t$$

- $s_t \leq q_t \leq q_t^*$:

$$q_{t+1} = q_t + \eta_t(-\alpha) + \eta_t(\frac{|s_t - q_t|}{2B}h(c*(s_t - q_t)) \pm \epsilon\left((1 - \frac{|s_t - q_t^*|}{2B})h(c*(s_t - q_t^*))\right))$$

$$\geq q_t + \eta_t(-\alpha) - \eta_t(1 - \epsilon)$$
$$\geq -(1 + \alpha - \epsilon)M_t$$

**Proposition 2** *Under Assumption 1 and Proposition 1 and $\eta_t \leq \frac{B}{2-\alpha-\epsilon}$, , we have bounds for two feedback:*

$$|s_t - q_t| \leq B + (2 - \alpha - \epsilon)M_{t-1}, \qquad \text{for any } t \geq 1,$$
$$|s_t - q_t^*| \leq 2B, \qquad \text{for any } t \geq 1.$$

*Proof.* Based on $s_t \in [0, B]$ and Proposition 1, under the assumption that $\eta \leq \frac{B}{2-\alpha-\epsilon}$ and $\epsilon \geq 0$,

$$|s_t - q_t| = \max\{q_t - s_t, s_t - q_t\} \leq \max\{q_t, s_t - q_t\}$$
$$\leq \max\{B + (2 - \alpha - \epsilon)M_{t-1},\ B + (1 + \alpha - \epsilon)M_{t-1}\}$$
$$\leq B + (2 - \alpha - \epsilon)M_{t-1}.$$

Hence, $|s_t - q_t| \leq c[B + (2 - \alpha - \epsilon)M_{t-1}]$.

**Theorem 1** *Let $\{\eta_t\}_{t \geq 1}$ be an arbitrary positive sequence. Under Assumption 1 and $\eta_t \leq \frac{2B}{2-\alpha-\epsilon}$, the prediction interval generated by Equation 10 with proportional learning rate $\eta_t$ satisfies:*

$$\left|\frac{1}{T}\sum_{t=1}^{T}(err_t - \alpha)\right| \leq \frac{(B + M_{T-1})\|\Delta_{1:T}\|_1}{T} + \frac{(1-\epsilon)B + (2-\alpha-\epsilon)M_{T-1}}{2B}, \qquad (12)$$

*where $\|\Delta_{1:T}\|_1 = |\eta_1^{-1}| + \sum_{t=2}^{T}|\eta_t^{-1} - \eta_{t-1}^{-1}|$, $M_T = \max_{1 \leq r \leq T}\eta_r$.*

*Proof of Theorem 1.*

$$\left|\frac{1}{T}\sum_{t=1}^{T}(\text{err}_t - \alpha)\right| = \left|\frac{1}{T}\sum_{t=1}^{T}\left(\sum_{r=1}^{t}\Delta_r\right) \cdot \eta_t(\text{err}_t - \alpha)\right|$$

$$= \left|\frac{1}{T}\sum_{r=1}^{T}\Delta_r\left(\sum_{t=r}^{T}\eta_t(\text{err}_t - \alpha)\right)\right|$$

$$= \left|\frac{1}{T}\sum_{r=1}^{T}\Delta_r\left(\sum_{t=r}^{T}\left[q_{t+1} - q_t - \eta_t(\frac{|s_t - q_t|}{2B}h(s_t - q_t) - \epsilon\frac{|s_t - q_t^*|}{2B}h(s_t - q_t^*))\right]\right)\right|$$

$$\leq \left|\frac{1}{T}\sum_{r=1}^{T}\Delta_r(q_{T+1} - q_r)\right| + \left|\frac{1}{T}\sum_{r=1}^{T}\Delta_r\sum_{t=r}^{T}\eta_t\frac{|s_t - q_t|}{2B}h(s_t - q_t) - \epsilon\frac{|s_t - q_t^*|}{2B}h(s_t - q_t^*))\right|$$

$$\leq \left|\frac{1}{T}\sum_{r=1}^{T}\Delta_r(q_{T+1} - q_r)\right| + \left|\frac{1}{T}\sum_{t=1}^{T}(\frac{|s_t - q_t|}{2B}h(s_t - q_t) - \epsilon\frac{|s_t - q_t^*|}{2B}h(s_t - q_t^*))\right|$$

$$\leq \left|\frac{1}{T}\sum_{r=1}^{T}\Delta_r(q_{T+1} - q_r)\right| + \frac{(1 - 2\epsilon)B + (2 - \alpha - \epsilon)M_{T-1}}{2B}$$

$$\leq \frac{(B + M_{T-1})\|\Delta_{1:T}\|_1}{T} + \frac{(1 - 2\epsilon)B + (2 - \alpha - \epsilon)M_{T-1}}{2B}$$

# B ADDITIONAL EXPERIMENTAL RESULTS

## B.1 DESCRIPTION OF DATASETS

- **Amazon and Google**: contain thirty blue-chip stock prices, with daily opening prices forecasted on a log scale, spanning the period from January 1, 2006 to December 31, 2014.

- **Microsoft**: includes daily stock opening prices from April 1, 2015 to May 31, 2021.

- **Delhi temperature**: provides daily measurements of temperature (averaged over eight readings taken at three-hour intervals), humidity, wind speed, and atmospheric pressure in Delhi, from January 1, 2003 to April 24, 2017, collected via the Weather Underground API.

- **Electricity demand**: records electricity demand in New South Wales at half-hour intervals from May 7, 1996 to December 5, 1998.

## B.2 EXPERIMENTAL RESULTS

### B.2.1 ELECTRICITY DEMAND

For the electricity demand dataset (Table 6), the performance of the DDCI variants is broadly comparable to that of the ECI baselines rather than superior. Across all forecasting models, DDCI and DDCI-Nex achieve interval widths that are similar to the best ECI methods while maintaining the target coverage level. Although the improvements observed in the stock datasets are not replicated here, the results indicate that DDCI remains competitive without loss of validity. This suggests that the performance of DDCI in the electricity demand setting remains stable and consistent with strong baseline approaches.

Table 6: The experimental results in the electricity demand dataset.

| Method | Prophet Model | | | AR Model | | | Theta Model | | | Transformer | | |
|---|---|---|---|---|---|---|---|---|---|---|---|---|
| | Coverage (%) | Avg. width | Median width | Coverage (%) | Avg. width | Median width | Coverage (%) | Avg. width | Median width | Coverage (%) | Avg. width | Median width |
| ACI | 90.1 | $\infty$ | 0.443 | 90.1 | $\infty$ | 0.105 | 90.2 | $\infty$ | 0.055 | 90.2 | $\infty$ | 0.109 |
| OGD | 89.8 | 0.433 | 0.435 | 90.0 | 0.133 | 0.115 | 90.1 | 0.081 | 0.075 | 90.1 | 0.139 | 0.120 |
| SF-OGD | 89.9 | 0.419 | 0.426 | 90.0 | 0.129 | 0.116 | 90.3 | 0.106 | 0.095 | 90.3 | 0.141 | 0.114 |
| decay-OGD | 90.1 | 0.531 | 0.521 | 90.1 | 0.122 | 0.099 | 90.0 | 0.100 | 0.059 | 90.3 | 0.147 | 0.111 |
| PID | 90.1 | **0.207** | **0.177** | 90.0 | 0.434 | 0.432 | 89.9 | 0.413 | 0.411 | 89.9 | 0.428 | 0.435 |
| ECI | 90.0 | 0.395 | 0.382 | 90.0 | 0.118 | 0.098 | 89.9 | **0.071** | 0.055 | 90.2 | 0.135 | 0.111 |
| ECI-cutoff | 90.0 | 0.405 | 0.396 | 90.0 | **0.117** | **0.096** | 90.1 | 0.072 | 0.055 | 90.2 | **0.133** | **0.108** |
| ECI-integral | 90.1 | 0.420 | 0.398 | 90.0 | 0.118 | 0.098 | 89.9 | 0.072 | 0.055 | 90.2 | 0.135 | 0.111 |
| **DDCI** | 90.1 | 0.392 | 0.390 | 90.0 | 0.118 | 0.099 | 89.9 | 0.072 | **0.054** | 90.2 | 0.135 | 0.111 |
| **DDCI-Nex** | 90.2 | 0.402 | 0.398 | 90.0 | 0.118 | 0.099 | 89.9 | 0.072 | **0.054** | 90.2 | 0.135 | 0.111 |

### B.2.2 VARYING SIGNIFICANCE LEVELS

We conducted experiments on AMZN stock dataset under Prophet model with different significance level (0.2, 0.1 and 0.05) to show the performance of our proposed method. Our proposed method keeps the best performance compared to all baselines. Details are given in the Table 7:

Table 7: Results under different significance levels $\alpha$

| Method | $\alpha = 0.20$ | | | $\alpha = 0.10$ | | | $\alpha = 0.05$ | | |
|---|---|---|---|---|---|---|---|---|---|
| | Coverage (%) | Avg. width | Winkler Score | Coverage (%) | Avg. width | Winkler Score | Coverage (%) | Avg. width | Winkler Score |
| OGD | 79.7 | 33.70 | 44.27 | 89.6 | 55.15 | 67.57 | 94.3 | 77.24 | 92.63 |
| SF-OGD | 79.7 | 44.51 | 58.31 | 89.5 | 61.47 | 73.94 | 94.5 | 79.11 | 92.74 |
| decay-OGD | 80.1 | 46.92 | 57.98 | 89.9 | 97.22 | 107.79 | 90.9 | 107.32 | 159.64 |
| PID | 80.3 | 36.45 | 47.97 | 89.8 | 52.56 | 56.02 | 94.8 | 54.65 | 66.58 |
| ECI | 80.8 | 35.39 | 45.85 | 90.1 | 47.00 | 59.74 | 94.8 | 60.10 | 73.81 |
| ECI-cutoff | 80.0 | 31.81 | 40.93 | 89.7 | 43.46 | 53.21 | 94.8 | 55.97 | 66.65 |
| ECI-integral | 80.0 | 32.21 | 41.26 | 89.8 | 42.01 | 51.85 | 94.8 | 56.00 | 67.19 |
| **DDCI** | 79.9 | 24.91 | 33.76 | 89.7 | **30.87** | 41.05 | 94.8 | 41.17 | 51.56 |
| **DDCI-Nex** | 79.8 | 25.25 | 34.27 | 89.7 | 31.28 | 40.96 | 94.7 | **41.09** | **51.32** |
| **DDCI-Softsign** | 80.1 | **24.56** | **33.34** | 89.7 | 31.11 | **40.90** | 94.8 | 41.23 | 51.77 |

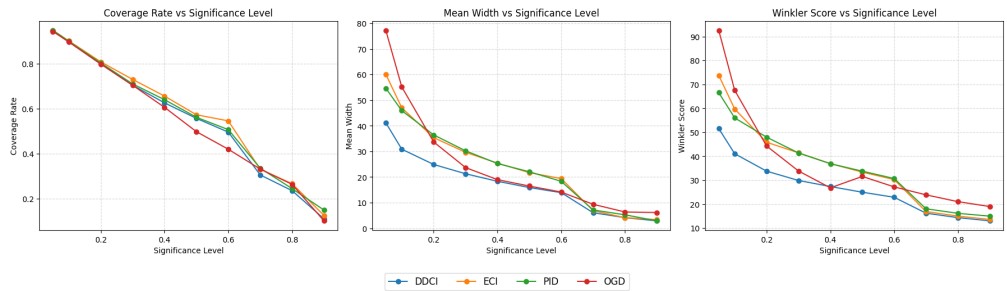

Figure 6: Comparison of empirical coverage rate, mean interval width, and Winkler Score across a wide range of significance levels ($\alpha = 0.05$–0.9) for four online conformal inference methods: DDCI, ECI, PID, and OGD.

To comprehensively evaluate the calibration behaviour of the online conformal methods, we conducted a multi-level significance experiment by sweeping $\alpha$ from 0.05 to 0.9. For each level, we measured the empirical coverage, the mean interval width, and the Winkler score across four methods (DDCI, ECI, PID and OGD).

Across the full range of significance levels from 0.05 to 0.9, DDCI consistently demonstrates superior calibration stability and interval efficiency compared with existing online conformal methods. In terms of efficiency, DDCI achieves the narrowest mean interval widths across almost all $\alpha$, reflecting its ability to maintain reliable calibration without resorting to excessively conservative intervals. This advantage is further reinforced by the Winkler Score results: DDCI attains the lowest Winkler Score throughout the entire spectrum, indicating the most favourable balance between coverage accuracy and interval compactness. Overall, the empirical curves of coverage, width, and Winkler Score collectively show that DDCI offers a more robust and efficient update rule, yielding both tighter and better-calibrated prediction intervals under varying levels of uncertainty.

### B.2.3 LEARNING RATE ANALYSIS

In this section, we analyse the impact of different learning rates on the performance of the proposed method, using the Google stock dataset as an example. The learning rate range follows Wu et al. (2025). For PID, ECI, and DDCI, the learning rates considered are [1, 0.5, 0.1, 0.05]. The results for coverage and prediction set of four underlying algorithms are given from Fig. 7 to Fig. 14. It is obvious that our proposed method loses validity and generates extremely wide prediction intervals when the learning rate is set to 1, which is consistent with our theoretical analysis. However, such large learning rate always yields conservative prediction intervals for all online conformal methods.

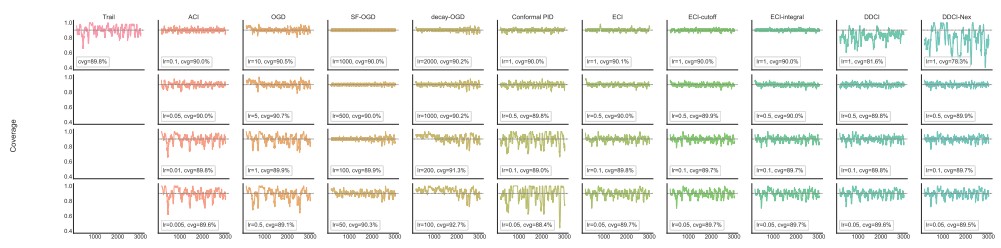

Figure 7: Coverage result for AR model.

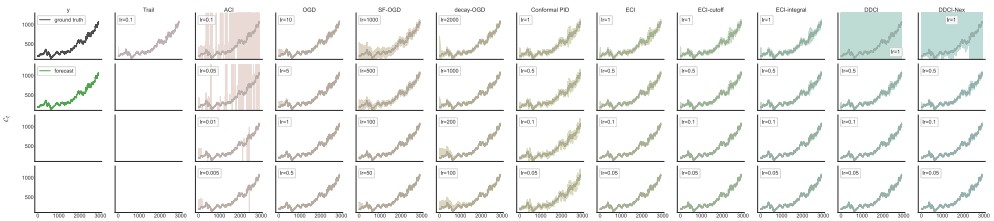

Figure 8: Prediction sets result for AR model.

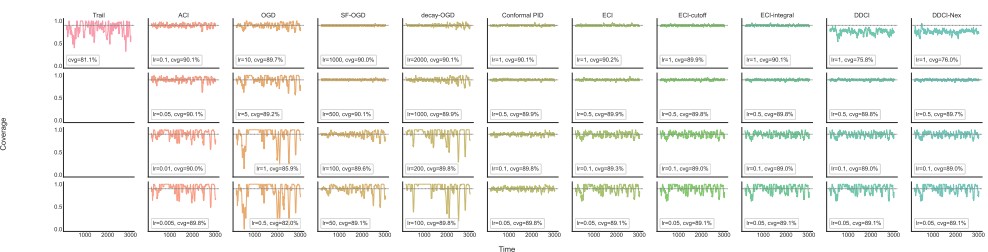

Figure 9: Coverage result for Prophet model.

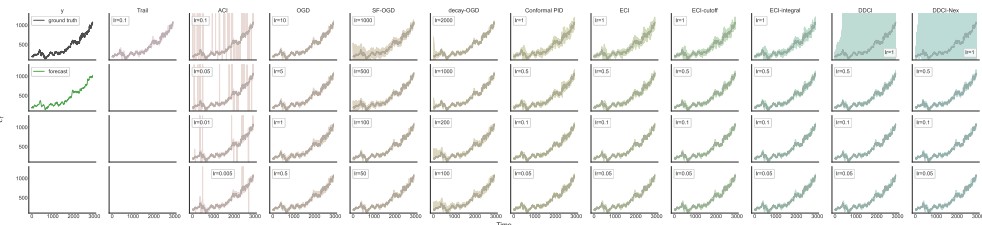

Figure 10: Prediction sets result for Prophet model.

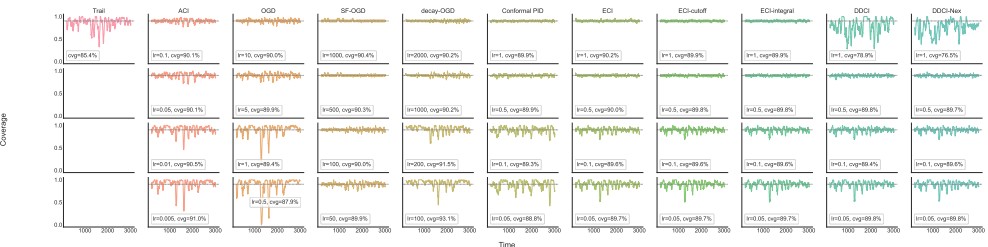

Figure 11: Coverage result for Theta model.

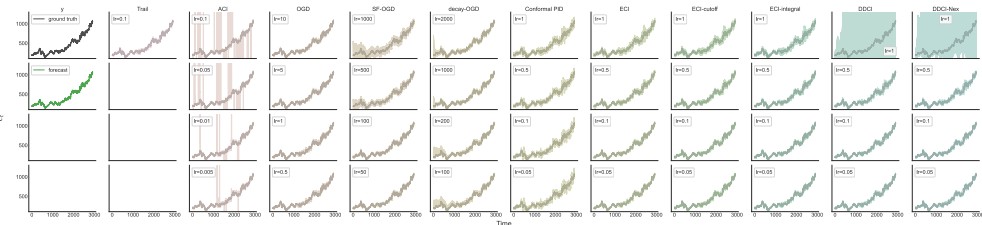

Figure 12: Prediction sets result for Theta model.

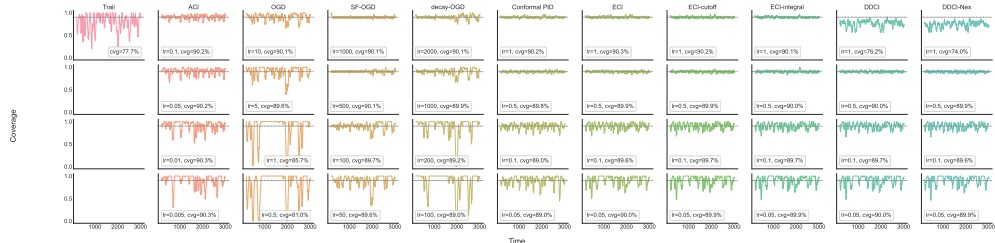

Figure 13: Coverage result for Transformer model.

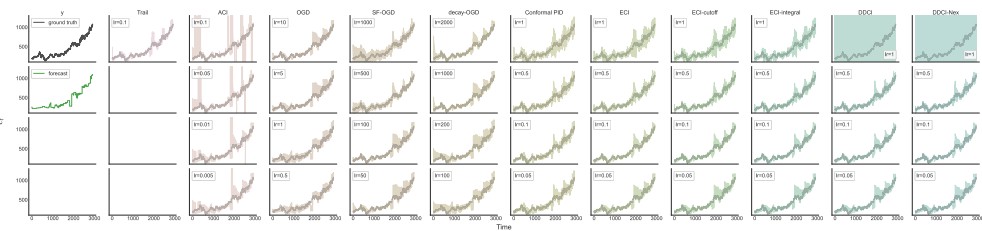

Figure 14: Prediction sets result for Transformer model.

## C    CODE LINK

Here is the anonymous link of the code to reproduce the experimental results and figures:
`https://anonymous.4open.science/r/DDCI-47A7`

## D    USAGE OF LANGUAGE MODELS

Large Language Models were used to polish and improve the grammar, spelling, and readability of this manuscript, but all research design, analysis, and interpretation were conducted by the authors.