# OpenReview forum: "Dynamic Dual-Feedback Conformal Inference for Time series Forecasting"
_ICLR.cc/2026/Conference — Submitted to ICLR 2026_

### Official Review · Reviewer_dyja · 2025-10-28

**Soundness:** 2
**Presentation:** 3
**Contribution:** 2
**Rating:** 4
**Confidence:** 5

**Summary:**

This paper introduces Dynamic Dual-feedback Conformal Inference (DDCI), a new online conformal prediction method for time series forecasting. DDCI enhances standard adaptive conformal inference by incorporating two feedback signals: actual coverage error and an estimated threshold from conformal inference. This dual-feedback design stabilizes the update process, producing narrower prediction intervals while maintaining target coverage, as validated theoretically and empirically across multiple datasets and forecasting models.

**Strengths:**

1. The idea of incorporating estimated feedback from conformal inference into the online update process is well-motivated.
 2. The mathematical proofs are easy to follow and the notations are clear.
 3. The experiment section is well-presented.

**Weaknesses:**

1. A recurring issue in this line of work, including the present paper, is the treatment of parameters. While this work introduces an additional parameter, a more thoughtful approach to parameter selection would have strengthened the contribution.
 2. Although DDCI shows significant improvements on stock datasets, the gains on the Delhi temperature and electricity demand datasets are less pronounced. This suggests that the method may be more suited to certain types of non-stationarity, and a deeper analysis of its limitations across different time series characteristics would be beneficial.
 3. See questions.

**Questions:**

1. Although it is reasonable to utilize the estimated threshold to stablize and counterbalance the intervals, is there any theoretical intuition on why choosing the estimated feedback in the form of (5), or theoretical results on the impact of added feedback terms compared to OGD? For example, why not simply take the estimated feedback function as $-|x|/2B*tanh(cx)$?
2. The experimental results of ECI with AR model seems consistently outperforms DDCI (except for Microsoft data). How do the authors comment about this?
3. Since the main contribution lies in the estimated threshold part, can you provide a more specific way on how to choose $\epsilon$?

---

> ### Author Response · Authors · 2025-11-21
> **Rebuttal to Reviewer dyja (1/2)**
>
> Dear Reviewer dyja,  We sincerely appreciate the time and effort you devoted, as well as your helpful suggestions. Our responses to your concerns are given below. Please feel free to let us know if anything requires further clarification.
>
> >W1: Additional parameter
>
> We understand your concern about parameters. We agree that parameter selection is an important aspect of online conformal methods, as it is a trade-off between predictive efficiency and update variability.
>
> We would like to clarify that in our Figure 3, the observed sensitivity to $\epsilon$ is not primarily driven by itself. The differences arise because each base model (AR, Prophet, Theta, Transformer) requires a distinct learning rate (0.05, 0.5, 0.1, 0.5 respectively), following the settings in ECI. Across all base models, smaller $\epsilon$ consistently reduce interval widths by preventing unnecessary expansions induced by the quantile-loss term in Eq. 3.
>
> >W2: Explanation for the performance on Delhi/Electricity and certain type of non-stationary.
>
> Thank you for this question. Our method relies on the estimated threshold $q_t^\*$ from conformal prediction to counterbalance the interval update process. Because of this design, the estimated feedback may have a side effect in cases where $q_t^\*$ represents conservative uncertainty at time $t$. In such situations, the deviation $e_t^\*$ can become relatively large, leading to unnecessarily wider intervals. For example, on the Delhi dataset using the Prophet model, the original conformal prediction already attains coverage of 90.50\% with an average width of 9.25, which means $q_t^\*$ is less efficient. In this case, the estimated feedback term may enlarge the interval due to the large deviation $e_t^\*$.
>
> For certain type of non-stationary, we constructed three datasets exhibiting different forms of non-stationarity:
> * Random Walk Trend: $X_t = X_{t-1} + \sigma_t \eta_t,\ \eta_t \sim \mathcal{N}(0,1)$, where $\sigma_t = \sigma_0 \left(1 + \kappa |\sin(2\pi t / P)|\right)$. This introduces smooth drift and cyclical volatility changes.
> * Exponential Trend: $X_t = X_{t-1}\exp(\mu + \sigma \varepsilon_t),\ \varepsilon_t \sim \mathcal{N}(0,1)$.
> * Changepoint Process (a piecewise log-linear model with two regime shifts at $c_1 = 600$ and $c_2 = 1200$): $\log X_t = \log X_{t-1} + \mu_{(t)} + \sigma_{(t)}\varepsilon_t$, where $\varepsilon_t \sim \mathcal{N}(0,1)$ and $(\mu_{(t)},\sigma_{(t)})$ change across segments.
>
> Across all three settings, DDCI consistently achieves the best overall performance, offering both stable coverage and the most informative intervals.. The results are averaged over 5 runs and summarised in Table below:
>
> | Method | Setting 1 Cov. (%) | Avg. width | Winkler Score | Setting 2 Cov. (%) | Avg. width | Winkler Score | Setting 3 Cov. (%) | Avg. width | Winkler Score |
> |------------|----------------------------|------------|----------------|-----------------------------|------------|----------------|-----------------------------|------------|----------------|
> | OGD | 90.0 ± 0.12 | 11.01 ± 0.50 | 12.82 ± 0.54 | 89.9 ± 0.17 | 17.28 ± 7.59 | 21.01 ± 8.98 | 85.0 ± 4.78 | 371.43 ± 350.57 | 506.07 ± 541.8 |
> | SF-OGD | 90.7 ± 0.16 | 75.75 ± 0.11 | 84.93 ± 0.14 | 90.3 ± 0.23 | 76.83 ± 1.50 | 86.57 ± 2.44 | 89.4 ± 0.98 | 183.13 ± 138.25 | 213.06 ± 162.6 |
> | decay-OGD | 89.7 ± 0.82 | 21.34 ± 2.10 | 23.32 ± 2.44 | 84.3 ± 4.90 | 44.05 ± 34.24 | 56.04 ± 47.25 | 59.6 ± 22.53 | 419.60 ± 307.13 | 2202.6 ± 2148.2 |
> | PID | 89.9 ± 0.11 | 13.25 ± 1.18 | 15.19 ± 1.33 | 89.8 ± 0.08 | 18.04 ± 6.53 | 21.72 ± 7.95 | 89.8 ± 0.13 | 124.17 ± 100.70 | 144.22 ± 115.73 |
> | ECI | 90.0 ± 0.10 | 13.07 ± 1.39 | 15.07 ± 1.53 | 90.0 ± 0.12 | 19.95 ± 7.15 | 30.58 ± 11.92 | 81.9 ± 17.84 | 88.85 ± 54.96 | 119.52 ± 61.22 |
> | ECI-cutoff | 89.9 ± 0.07 | 10.64 ± 1.06 | 12.50 ± 1.21 | 90.0 ± 0.09 | 17.57 ± 7.23 | 27.63 ± 11.52 | 89.8 ± 0.11 | 123.54 ± 100.01 | 146.74 ± 116.99 |
> | ECI-integral | 90.0 ± 0.06 | 10.90 ± 1.06 | 12.82 ± 1.27 | 90.0 ± 0.16 | 18.00 ± 7.80 | 27.87 ± 11.89 | 89.8 ± 0.07 | 125.46 ± 101.28 | 148.14 ± 117.75 |
> | **DDCI** | 89.9 ± 0.02 | **8.08 ± 0.82** | **9.82 ± 1.04** | 90.0 ± 0.19 | **13.15 ± 5.35** | **22.93 ± 9.78** | 89.8 ± 0.10 | **91.36 ± 72.59** | **111.90 ± 84.47** |
> | **DDCI-Nex** | 89.9 ± 0.03 | 8.35 ± 0.84 | 10.07 ± 1.00 | 89.9 ± 0.14 | 15.36 ± 6.75 | 25.20 ± 11.18 | 89.8 ± 0.10 | 109.52 ± 89.26 | 131.36 ± 104.59 |
>
> DDCI consistently produces narrower intervals while maintaining coverage comparable to baseline methods, particularly in settings with stronger non-stationarity (e.g., changepoints or stochastic trends). This suggests that the dual-feedback mechanism is especially effective when the score distribution exhibits abrupt or irregular shifts.

---

> > ### Author Response · Authors · 2025-11-21
> > **Rebuttal to Reviewer dyja (2/2)**
> >
> > >Q1: The form of estiamted feedback in Eq. (5).
> >
> > In our method, this term is intended to stabilize and counterbalance the intervals, and its specific form is motivated by two theoretical considerations.
> >
> > First, for small deviations $|e_t^\*|$ (i.e., when the score is close to the estimated conformal threshold), the function $f(e_t^\*) = (1 - \frac{|e_t^\*|}{2B})\tanh(c e_t^\*)$ satisfies $\tanh(c e_t^\*) \approx c e_t^\*$, and therefore $f(e_t^\*) \approx c e_t^\*$ when it is small. This means that the estimated feedback behaves as a linear restoring force around the target quantile, producing a contracting update that corrects small mis-calibrations smoothly and avoids the oscillatory behaviour observed in existing methods.
> >
> > Second, for large $|e_t^\*|$, we have $\tanh(c e_t^\*) \approx sign(e_t^\*)$, so that $f(e_t^\*) \approx (1 - \frac{|e_t^\*|}{2B}) sign(e_t^\*),$ which gradually attenuates to zero as $|e_t^\*| \to 2B$. This produces a redescending behaviour: extreme deviations are down-weighted so that the estimated feedback does not cause unnecessary expansions of the prediction interval, thereby improving efficiency without compromising stability.
> >
> > For these reasons, the proposed form in Eq. 5 is not merely heuristic, but is designed to provide a linear stabilising effect near the target level.
> >
> > >Q2: The performance under AR model.
> >
> > The answer to this question is similar to our response to W2. The behaviour of DDCI under the AR model is closely related to the strength of the conformal threshold $q_t^\*$. On the Climate dataset, for example, the original conformal prediction (using AR model) already achieves coverage of 89.66\% with an average width of 8.32, which indicates that the conformal thresholds are relatively conservative. In such cases, the estimated feedback may amplify the deviation $e_t^*$, leading to wider intervals and reducing the relative advantage of DDCI.
> >
> > >Q3: How to choose $\epsilon$?
> >
> > We thank the reviewer for this helpful suggestion. In our framework, the hyperparameter $\epsilon$ controls the relative influence of the estimated feedback, and its suitable value is closely related to the degree of non-stationarity present in the underlying score sequence. In principle, the empirical coverage rate of conformal prediction from past window can reflect the degree of non-stationary and this information can be used to set $\epsilon$ dynamically.

---

### Official Review · Reviewer_4pSB · 2025-10-30

**Soundness:** 2
**Presentation:** 3
**Contribution:** 2
**Rating:** 4
**Confidence:** 3

**Summary:**

This paper proposes a new online conformal prediction method by adding an "Estimated Feedback" to the ECI method, which incorporates more information from historical data. The experiment results show the superiority of the proposed method over the existing baselines.

**Strengths:**

The estimated feedback is new in the online CP, which helps construct more stable and potentially shorter prediction intervals.

**Weaknesses:**

**1. The new method introduces more tuning parameters.**

Compared with ECI, there is an additional parameter $\epsilon$ in the update rule. According to Figure 3, the size is very sensitive to the choice of $\epsilon$. However, choosing $\epsilon$ is challenging in the online task.

**2. About the role of the estimated feedback**.

The estimated feedback has an opposite sign to the actual feedback, intending to smooth the update. To achieve this goal, we can choose any positive sequence to replace $h(c e_t^* \ )$, e.g., $h(c e_t)$. Why do we use $e_t^*\ $?

 Also, the statement in Lines 213-214 seems incorrect. Even though $q_t^* \ $ is the true $1-\alpha$ quantile of  $s_t$, the difference $e_t^* = |s_t - q_t^*|$ can be very large.

Overall, the role of estimated feedback is not well discussed, which makes it like a tuning trick. I suggest the authors provide a fundamental explanation based on a working data model.

**Questions:**

1. The actual feedback in (4) is not consistent with that in ECI.

2. What if we only use the estimated feedback $h(c e_t^* \ )$ to replace the actual feedback in ECI.

---

> ### Author Response · Authors · 2025-11-21
> **Rebuttal to Reviewer 4pSB**
>
> Dear Reviewer 4pSB, We sincerely appreciate the time and effort you devoted, as well as your helpful suggestions. We have addressed all of your concerns; please feel free to let us know if anything requires further clarification.
>
> >W1: Sensitivity to $\epsilon$.
>
> We agree that the introduction of $\epsilon$ raises questions about tuning complexity in an online setting. However, the apparent sensitivity shown in Fig. 3 is not primarily driven by $\epsilon$ itself. The different trends across panels arise because each base model (AR, Prophet, Theta, Transformer) employs a distinct learning rate (0.05, 0.5, 0.1, 0.5, respectively), following the experimental setup of ECI. This causes the scale of the score sequence $\{s_t\}$, and consequently the update magnitude, to differ substantially across models. Across all architectures, a consistent pattern emerges: smaller values of $\epsilon$ lead to narrower intervals because they moderate the influence of the quantile-loss term in Eq. (3).
>
> >W2: About the role of the estimated feedback.
>
> Our use of $h(c e_t^\*)$ (rather than an arbitrary positive sequence or $h(c e_t)$) is due to the role of the conformal estimated threshold. Since $q_t^\*$ is based on historical residuals, $e_t^\*$ is less noisy than the instantaneous deviation $e_t$, producing stable feedback. Moreover, conformal calibration is defined relative to $q_t^\*$, even though validity may fail under non-exchangeability. Using $e_t^\*$ therefore ensures that the smoothing operates around the correct conformal reference. For these reasons, $h(c e_t^\*)$ provides both statistical coherence and greater stability than alternatives.
>
> The estimated feedback is not intended as a tuning trick. Its form is justified by two complementary considerations.
>
> First, for small deviations $|e_t^\*|$ (i.e., when the score is close to the estimated conformal threshold), the function $f(e_t^\*) = (1 - \frac{|e_t^\*|}{2B})\tanh(c e_t^\*)$ satisfies $\tanh(c e_t^\*) \approx c e_t^\*$, and therefore $f(e_t^\*) \approx c e_t^\*$ when it is small. This means that the estimated feedback behaves as a linear restoring force around the target quantile, producing a contracting update that corrects small mis-calibrations smoothly and avoids the oscillatory behaviour observed in existing methods.
>
> Second, for large $|e_t^\*|$, we have $\tanh(c e_t^\*) \approx sign(e_t^\*)$, so that $f(e_t^\*) \approx (1 - \frac{|e_t^\*|}{2B}) sign(e_t^\*),$ which gradually attenuates to zero as $|e_t^\*| \to 2B$. This produces a redescending behaviour: extreme deviations are down-weighted so that the estimated feedback does not cause unnecessary expansions of the prediction interval, thereby improving efficiency without compromising stability.
>
> Overall, these two properties show that the form of the estimated feedback in Eq. 5 is not heuristic but is designed to provide linear stabilisation near the target quantile and robustness against large deviations.
>
> >Q1: The actual feedback in (4) is not consistent with that in ECI.
>
> The apparent inconsistency comes from the fact that our method and ECI are based on different design principles. In ECI, the actual feedback term is explicitly constructed to limit the impact of outliers: when $e_t$ is large, the update rule is designed to restrict the amount of interval expansion. In contrast, DDCI achieves this control through an estimated feedback term that suppresses unnecessary interval enlargement when large deviations occur. This is a key conceptual distinction between the two methods.
>
> Moreover, ECI requires an additional assumption on the form of the actual feedback function to guarantee stability, while DDCI does not rely on such an assumption. Our update rule is designed to remain stable even without imposing constraints on the structure of the feedback term.
>
> >Q2: What if we only use the estimated feedback $h(ce^\*)$ to replace the actual feedback in ECI.
>
> Using only the estimated feedback $h(c e_t^\*)$ to replace the actual feedback in ECI would not affect the validity, since the coverage guarantee is achieved by the quantile-loss term. However, it would reduce predictive efficiency. The oscillatory behaviour in ECI is driven by the quantile-loss term, which reacts strongly when the true label falls outside the interval. The actual feedback $e_t$ is therefore essential for controlling these large expansions and ensuring that the update responds promptly to abrupt changes in the residuals. Therefore, it is needed to use the actual feedback to mitigate the large expansion. And, under distribution shift in nonexchangeable settings, the estimated threshold $q_t^\*$ may become misaligned with the current score distribution and thus cannot serve as a reliable sole reference for the interval update process. For this reason, DDCI uses the estimated feedback as a stabilising complement rather than a replacement for the actual feedback.

---

### Official Review · Reviewer_CFC4 · 2025-10-30

**Soundness:** 3
**Presentation:** 3
**Contribution:** 2
**Rating:** 6
**Confidence:** 3

**Summary:**

This paper proposes Dynamic Dual-feedback Conformal Inference (DDCI), a new online conformal prediction method for time series forecasting. DDCI implements the control-theory concept of dual feedback, which incorporates two feedback signals: (1) error / miscoverage - normalized and squashed error magnitude, and (2) a stabilizing signal derived from empirical quantiles of previous scores.

**Strengths:**

- The paper is clearly written and well organized. I understood the method and the experiment well.

- The dual-feedback idea is nice and a nice fit for dynamic control of coverage. Some adaptability is sacrificed for stability, but the stability of quantile updates results in more efficient intervals.

- The tasks used in the experiment section is diverse and the baseline selection is very up-to-date. DDCI achieves coverage and sharpness on all datasets, verifying the theory.

- I think the experiment section is overall quite thoughtful. The natural extension to weighted quantiles (DDCI-Nex) is reasonable and well-explored, and the sensitivity analysis on epsilon is explained clearly.

**Weaknesses:**

The flip side of spending 3.5 pages on comprehensive experiments is the lack of depth on theoretical discussions. As the dual-feedback concept is clearly inspired by control theory (fast and slow feedback loops), the paper would benefit for more detailed analysis on adaptation speed, coverage stability, computational cost, etc explicitly compared to ACI (pure P control) and Conformal PID. Specifically:

- How is dual-feedback related to PID's three terms (P, I, D)? Could DDCI be expressed as a PID controller with specific tuning?
- Are there advantages to dual-feedback over PID, or vice versa? Under what situations would PID control outperform DDCI and vice versa?
- Is there a specific setup where you can prove that the DDCI has smaller interval than other methods? by what magnitude? what does this magnitude depend on?
- Can these concepts guide the setting of the hyperparameters such as $\epsilon$?

**Questions:**

See weakness.

---

> ### Author Response · Authors · 2025-11-21
> **Rebuttal to Reviewer CFC4**
>
> Dear Reviewer CFC4, We sincerely appreciate the time and effort you devoted, as well as your helpful suggestions. We have addressed all of your concerns; please feel free to let us know if anything requires further clarification.
>
> >Q1: Relation to PID.
>
> Although both PID and DDCI use quantile loss, their feedback structures differ fundamentally. In conformal PID, the “I’’ term accumulates past coverage errors, and the “D’’ term depends on a scorecaster to predict future error.
>
> DDCI's dual-feedback term compares the true label with the interval true and estimated boundaries, producing a correction that is not analogous to either integral or derivative control. Thus, DDCI cannot be reduced to a PID with particular hyperparameters; its feedback is conceptually distinct and designed to stabilise and counterbalance interval updates.
>
> >Q2: Comparison between PID and DDCI.
>
> The “D’’ term in conformal PID relies on an extra scorecaster to predict future errors. This can be beneficial when the scorecaster is much stronger, for example, in the Electricity dataset, where a robust Theta scorecaster compensates for Prophet’s weaknesses and improves interval quality. However, this comes with two drawbacks: (1) PID becomes sensitive to the scorecaster’s modelling accuracy, and (2) the extra model can introduce instability when residuals are noisy or weakly predictable.
>
> By contrast, DDCI requires no extra model; its estimated feedback provides a smoother and more stable update mechanism. As a result, DDCI performs better when residuals are irregular, rapidly changing, or difficult to forecast.
>
> >Q3: Specific setup for DDCI.
>
> To address your concern, we constructed three datasets exhibiting different forms of non-stationarity:
> * Random Walk Trend: $X_t = X_{t-1} + \sigma_t \eta_t,\ \eta_t \sim \mathcal{N}(0,1)$, where $\sigma_t = \sigma_0 \left(1 + \kappa |\sin(2\pi t / P)|\right)$. This introduces smooth drift and cyclical volatility changes.
> * Exponential Trend: $X_t = X_{t-1}\exp(\mu + \sigma \varepsilon_t),\ \varepsilon_t \sim \mathcal{N}(0,1)$.
> * Changepoint Process (a piecewise log-linear model with two regime shifts at $c_1 = 600$ and $c_2 = 1200$): $\log X_t = \log X_{t-1} + \mu_{(t)} + \sigma_{(t)}\varepsilon_t$, where $\varepsilon_t \sim \mathcal{N}(0,1)$ and $(\mu_{(t)},\sigma_{(t)})$ change across segments.
>
> Across all three settings, DDCI consistently achieves the best overall performance, offering both stable coverage and the most informative intervals.. The results are averaged over 5 runs and summarised in Table below:
>
> | Method | Setting 1 Cov. (%) | Avg. width | Winkler Score | Setting 2 Cov. (%) | Avg. width | Winkler Score | Setting 3 Cov. (%) | Avg. width | Winkler Score |
> |------------|----------------------------|------------|----------------|-----------------------------|------------|----------------|-----------------------------|------------|----------------|
> | OGD | 90.0 ± 0.12 | 11.01 ± 0.50 | 12.82 ± 0.54 | 89.9 ± 0.17 | 17.28 ± 7.59 | 21.01 ± 8.98 | 85.0 ± 4.78 | 371.43 ± 350.57 | 506.07 ± 541.8 |
> | SF-OGD | 90.7 ± 0.16 | 75.75 ± 0.11 | 84.93 ± 0.14 | 90.3 ± 0.23 | 76.83 ± 1.50 | 86.57 ± 2.44 | 89.4 ± 0.98 | 183.13 ± 138.25 | 213.06 ± 162.6 |
> | decay-OGD | 89.7 ± 0.82 | 21.34 ± 2.10 | 23.32 ± 2.44 | 84.3 ± 4.90 | 44.05 ± 34.24 | 56.04 ± 47.25 | 59.6 ± 22.53 | 419.60 ± 307.13 | 2202.6 ± 2148.2 |
> | PID | 89.9 ± 0.11 | 13.25 ± 1.18 | 15.19 ± 1.33 | 89.8 ± 0.08 | 18.04 ± 6.53 | 21.72 ± 7.95 | 89.8 ± 0.13 | 124.17 ± 100.70 | 144.22 ± 115.73 |
> | ECI | 90.0 ± 0.10 | 13.07 ± 1.39 | 15.07 ± 1.53 | 90.0 ± 0.12 | 19.95 ± 7.15 | 30.58 ± 11.92 | 81.9 ± 17.84 | 88.85 ± 54.96 | 119.52 ± 61.22 |
> | ECI-cutoff | 89.9 ± 0.07 | 10.64 ± 1.06 | 12.50 ± 1.21 | 90.0 ± 0.09 | 17.57 ± 7.23 | 27.63 ± 11.52 | 89.8 ± 0.11 | 123.54 ± 100.01 | 146.74 ± 116.99 |
> | ECI-integral | 90.0 ± 0.06 | 10.90 ± 1.06 | 12.82 ± 1.27 | 90.0 ± 0.16 | 18.00 ± 7.80 | 27.87 ± 11.89 | 89.8 ± 0.07 | 125.46 ± 101.28 | 148.14 ± 117.75 |
> | **DDCI** | 89.9 ± 0.02 | **8.08 ± 0.82** | **9.82 ± 1.04** | 90.0 ± 0.19 | **13.15 ± 5.35** | **22.93 ± 9.78** | 89.8 ± 0.10 | **91.36 ± 72.59** | **111.90 ± 84.47** |
> | **DDCI-Nex** | 89.9 ± 0.03 | 8.35 ± 0.84 | 10.07 ± 1.00 | 89.9 ± 0.14 | 15.36 ± 6.75 | 25.20 ± 11.18 | 89.8 ± 0.10 | 109.52 ± 89.26 | 131.36 ± 104.59 |
>
> >Q4: Setting of parameter.
>
> In our framework, the parameter $\epsilon$ controls the strength of the estimated-feedback term, and its suitable value depends on the level of non-stationarity in the score sequence. When the score distribution drifts quickly, a larger $\epsilon$ helps stabilise updates, while in more stable regimes a smaller $\epsilon$ avoids over-smoothing.
>
> A possible way is to track empirical coverage over a rolling window: deviations from the target level signal local drift and can be used to adjust $\epsilon$ dynamically so that the feedback strength matches the degree of non-stationarity in the data.

---

### Official Review · Reviewer_68ke · 2025-11-01

**Soundness:** 2
**Presentation:** 2
**Contribution:** 2
**Rating:** 2
**Confidence:** 4

**Summary:**

The work proposes a conformal method for time series forecasting. The proposed method employs a dual-feedback system, not previously used, to obtain conformal regions that are more efficient than existing baselines.

**Strengths:**

The update rule to adapt the conformal set provides a flexible mechanism. The experiments demonstrate the efficacy of the proposed methods in the chosen settings.

**Weaknesses:**

The presentation can be improved. Many times, the terminology just sounds a bit odd, or there is repetition of ideas already mentioned.

The novelty seems limited over the existing methods.

See "Questions*" below for more

**Questions:**

1. Line 037: Conformal Inference was introduced before the 2005 book cited here. Consider rephrasing the sentence or citing the earlier work from Vovk.

2. Line 072: Upper Limitation? Maybe the authors meant an upper bound here.

3. The first paragraph in the Related Work section seems unnecessary; the work is a conformal work, and the mentioned uncertainty methods appear not to be used as baselines.

4. Line 101-103: Rather than intuitively defining conformal prediction, it would be better to write it down formally as a definition later. Note that the section is about conformal inference under nonexchangeability, but the description starts with traditional conformal prediction.

5. Line 119: The term residual-aware seems odd. Did the authors mean residual-controlling or something?

6. How dependent is the performance on the squashing function? It appears the work only explored tanh function,

7. The target confidence is set as high as 0.9, which is okay. However, it is necessary to demonstrate the performance of the proposed method with varying significance levels. A calibration curve may be helpful in determining if the proposed method works effectively in all cases.

8. For all the experiments, only single-time statistics are provided. It would be helpful to have the experiments repeated across multiple splits or seeds and the standard deviation reported to get a clearer picture of the method's performance.

9. For many results, where DDCI performs better, there is no clear indication if the difference in width is sufficient in comparison to the deviations in reported coverage to know if DDCI is indeed a better method. Other metrics, such as Winkler Score, might also be helpful to understand the method better.

---

> ### Author Response · Authors · 2025-11-21
> **Rebuttal to Reviewer 68ke (1/2)**
>
> Dear Reviewer 68ke, Thank you for your review and suggestions. We conducted all experiments required and showed the results below.
>
> > Q1-Q4: Presentation issue
>
> Thanks for your suggestions. We will rephrase the sentences according to your comments.
>
> >Q5: The term "residual-aware".
>
> We refer to our feedback function as “residual-aware’’ because it incorporates the scaling factor $\frac{1}{2B}$ in Eq. 4 and Eq. 5 to modulate how much the interval expands or contracts during each update. This scaling explicitly accounts for the magnitude of residuals, which can vary substantially across datasets and across different base models. In contrast, the error-quantified term in ECI does not adjust for these differences, treating residuals of all scales uniformly. Our design therefore enables more stable and adaptive updates, particularly in settings where the scale of residuals is heterogeneous.
>
> >Q6: Squashing function.
>
> We added **Softsign** as another squashing function. The performance of this function is similar to that of the **tanh** function. The results are given in the following experiment-related questions. We also added the Winkler Score as a new evaluation metric.
>
> >Q7: Varying significance levels.
>
> We present the results for the AMZN stock dataset under the Prophet model, evaluated at significance levels of 0.20, 0.10, and 0.05. Our proposed method keeps the best performance compared to all baselines. Details are given in the Table below:
>
> | **Method** | **α=0.20 Coverage (%)** | **Avg. width** | **Winkler Score** | **α=0.10 Coverage (%)** | **Avg. width** | **Winkler Score** | **α=0.05 Coverage (%)** | **Avg. width** | **Winkler Score** |
> |-----------|--------------------------|----------------|--------------------|--------------------------|----------------|----------------------|----------------------------|----------------|----------------------|
> | OGD | 79.7 | 33.70 | 44.27 | 89.6 | 55.15 | 67.57 | 94.3 | 77.24 | 92.63 |
> | SF-OGD | 79.7 | 44.51 | 58.31 | 89.5 | 61.47 | 73.94 | 94.5 | 79.11 | 92.74 |
> | decay-OGD | 80.1 | 46.92 | 57.98 | 89.9 | 97.22 | 107.79 | 90.9 | 107.32 | 159.64 |
> | PID | 80.3 | 36.45 | 47.97 | 89.8 | 52.56 | 56.02 | 94.8 | 54.65 | 66.58 |
> | ECI | 80.8 | 35.39 | 45.85 | 90.1 | 47.00 | 59.74 | 94.8 | 60.10 | 73.81 |
> | ECI-cutoff | 80.0 | 31.81 | 40.93 | 89.7 | 43.46 | 53.21 | 94.8 | 55.97 | 66.65 |
> | ECI-integral | 80.0 | 32.21 | 41.26 | 89.8 | 42.01 | 51.85 | 94.8 | 56.00 | 67.19 |
> | **DDCI** | 79.9 | 24.91| 33.76| 89.7 | **30.87** | 41.05 | 94.8 | 41.17| 51.56 |
> | **DDCI-Nex** | 79.8 | 25.25 | 34.27 | 89.7 | 31.28 | 40.96 | 94.7 | **41.09** | **51.32** |
> | **DDCI-Softsign** | 80.1 | **24.56** | **33.34** | 89.7 | 31.11 | **40.90** | 94.8 | 41.23 | 51.77 |

---

> ### Author Response · Authors · 2025-11-21
> **Rebuttal to Reviewer 68ke (2/2)**
>
> >Q8: Single-time statistics.
>
> The online updating conformal methods do not require to train a model to generate prediction intervals, which use the residuals computed from base models. Therefore, we generated three synthetic datasets and simulated them under five random seeds to address your concern. These datasets exhibit different forms of non-stationarity:
> * Random Walk Trend: $X_t = X_{t-1} + \sigma_t \eta_t,\ \eta_t \sim \mathcal{N}(0,1)$, where $\sigma_t = \sigma_0 \left(1 + \kappa |\sin(2\pi t / P)|\right)$. This introduces smooth drift and cyclical volatility changes.
> * Exponential Trend (a multiplicative-growth process): $X_t = X_{t-1}\exp(\mu + \sigma \varepsilon_t),\ \varepsilon_t \sim \mathcal{N}(0,1)$.
> * Changepoint Process (a piecewise log-linear model with two regime shifts at $c_1 = 600$ and $c_2 = 1200$): $\log X_t = \log X_{t-1} + \mu_{(t)} + \sigma_{(t)}\varepsilon_t$, where $\varepsilon_t \sim \mathcal{N}(0,1)$ and $(\mu_{(t)},\sigma_{(t)})$ change across segments.
>
> Across all three settings, DDCI consistently achieves the best overall performance, offering both stable coverage and the most informative intervals.. The results are averaged over 5 runs and summarised in Table below:
>
> | **Method** | **Setting 1** Cov. (%) | Avg. width | Winkler Score | **Setting 2** Cov. (%) | Avg. width | Winkler Score | **Setting 3** Cov. (%) | Avg. width | Winkler Score |
> |------------|----------------------------|------------|----------------|-----------------------------|------------|----------------|-----------------------------|------------|----------------|
> | OGD | 90.0 ± 0.12 | 11.01 ± 0.50 | 12.82 ± 0.54 | 89.9 ± 0.17 | 17.28 ± 7.59 | 21.01 ± 8.98 | 85.0 ± 4.78 | 371.43 ± 350.57 | 506.07 ± 541.8 |
> | SF-OGD | 90.7 ± 0.16 | 75.75 ± 0.11 | 84.93 ± 0.14 | 90.3 ± 0.23 | 76.83 ± 1.50 | 86.57 ± 2.44 | 89.4 ± 0.98 | 183.13 ± 138.25 | 213.06 ± 162.6 |
> | decay-OGD | 89.7 ± 0.82 | 21.34 ± 2.10 | 23.32 ± 2.44 | 84.3 ± 4.90 | 44.05 ± 34.24 | 56.04 ± 47.25 | 59.6 ± 22.53 | 419.60 ± 307.13 | 2202.6 ± 2148.2 |
> | PID | 89.9 ± 0.11 | 13.25 ± 1.18 | 15.19 ± 1.33 | 89.8 ± 0.08 | 18.04 ± 6.53 | 21.72 ± 7.95 | 89.8 ± 0.13 | 124.17 ± 100.70 | 144.22 ± 115.73 |
> | ECI | 90.0 ± 0.10 | 13.07 ± 1.39 | 15.07 ± 1.53 | 90.0 ± 0.12 | 19.95 ± 7.15 | 30.58 ± 11.92 | 81.9 ± 17.84 | 88.85 ± 54.96 | 119.52 ± 61.22 |
> | ECI-cutoff | 89.9 ± 0.07 | 10.64 ± 1.06 | 12.50 ± 1.21 | 90.0 ± 0.09 | 17.57 ± 7.23 | 27.63 ± 11.52 | 89.8 ± 0.11 | 123.54 ± 100.01 | 146.74 ± 116.99 |
> | ECI-integral | 90.0 ± 0.06 | 10.90 ± 1.06 | 12.82 ± 1.27 | 90.0 ± 0.16 | 18.00 ± 7.80 | 27.87 ± 11.89 | 89.8 ± 0.07 | 125.46 ± 101.28 | 148.14 ± 117.75 |
> | **DDCI** | 89.9 ± 0.02 | **8.08 ± 0.82** | **9.82 ± 1.04** | 90.0 ± 0.19 | **13.15 ± 5.35** | **22.93 ± 9.78** | 89.8 ± 0.10 | **91.36 ± 72.59** | **111.90 ± 84.47** |
> | **DDCI-Nex** | 89.9 ± 0.03 | 8.35 ± 0.84 | 10.07 ± 1.00 | 89.9 ± 0.14 | 15.36 ± 6.75 | 25.20 ± 11.18 | 89.8 ± 0.10 | 109.52 ± 89.26 | 131.36 ± 104.59 |
> | **DDCI-Softsign** | 89.9 ± 0.03 | 8.40 ± 0.84 | 10.16 ± 1.06 | 90.0 ± 0.17 | 13.44 ± 5.41 | 23.21 ± 9.62 | 89.8 ± 0.12 | 91.48 ± 72.84 | 111.93 ± 87.58 |
>
> >Q9: Winkler Score:
>
> We take the results on Microsoft dataset as an example. As the Table shows, our proposed method also show advantages compared to the existing methods.
>
> | **Method** | **Prophet** Coverage (%) | Avg. width | Winkler Score | **AR** Coverage (%) | Avg. width | Winkler Score | **Theta** Coverage (%) | Avg. width | Winkler Score | **Transformer** Coverage (%) | Avg. width | Winkler Score |
> |------------|--------------------------|------------|----------------|----------------------|------------|----------------|-------------------------|------------|----------------|-------------------------------|------------|----------------|
> | OGD | 90.0 | 3.78 | 4.80 | 90.7 | 4.37 | 4.49 | 89.9 | 2.49 | 3.31 | 89.9 | 3.80 | 5.00 |
> | SF-OGD | 90.2 | 6.91 | 7.14 | 90.0 | 6.82 | 7.17 | 90.2 | 6.98 | 7.31 | 89.9 | 7.08 | 7.08 |
> | decay-OGD | 90.0 | 6.02 | 5.34 | 90.1 | 4.34 | 3.64 | 90.1 | 4.91 | 4.92 | 89.7 | 5.41 | 5.41 |
> | PID | 90.0 | 6.30 | 5.90 | 89.6 | 5.60 | 5.43 | 89.8 | 4.89 | 4.87 | 89.9 | 6.16 | 6.48 |
> | ECI | 90.4 | 4.85 | 5.24 | 90.2 | 3.76 | 4.20 | 90.2 | 2.48 | 3.43 | 89.9 | 4.35 | 5.02 |
> | ECI-cutoff | 89.8 | 4.05 | 4.59 | 89.8 | 3.04 | 3.58 | 89.8 | 2.44 | 3.39 | 89.9 | 4.08 | 4.87 |
> | ECI-integral | 89.9 | 4.22 | 4.71 | 90.2 | 3.67 | 4.15 | 90.2 | 2.46 | 3.42 | 89.9 | 4.13 | 4.98 |
> | **DDCI** | 89.8 | **3.26** | **3.71** | 89.8 | 2.93 | 3.58 | 89.8 | **2.39** | **3.31** | 89.9 | **3.34** | **4.10** |
> | **DDCI-Nex** | 89.8 | 3.40 | 3.89 | 89.8 | 2.92 | 3.50 | 89.9 | 2.41 | 3.33 | 89.9 | 3.44 | 4.23 |
> | **DDCI-Softsign** | 89.8 | 3.40 | 3.92 | 89.8 | **2.89** | **3.45** | 89.8 | **2.39** | 3.33 | 89.8 | 3.51 | 4.27 |

---

> > ### Comment · Reviewer_68ke · 2025-11-28
> >
> > Thanks for writing the rebuttal and for the clarifications.
> >
> > The method seems to be working well for the other significance levels (0.2 and 0.05), but I wish there were more levels there. Some conformal methods show better coverage when low significance levels are chosen, but not when they are higher. The stability of the method can be best assessed at higher levels, and hence I had asked for a calibration plot. Also, for the added significance levels, there are no error bars.
> >
> > "The online updating conformal methods do not require training a model to generate prediction intervals, which use the residuals computed from base models" - In such a situation, training different models or with different seeds and then testing the conformal method might have been a solution, am I right?

---

> > > ### Author Response · Authors · 2025-11-28
> > > **Further responses**
> > >
> > > Thank you for your clarification and suggestions. Our responses are given below:
> > >
> > > "Some conformal methods show better coverage when low significance levels are chosen" - This series of research (the baselines) is based on the control theory, where the validity is guaranteed by the quantile-loss function as Eq. (4). Therefore, the long-run coverage rate of these methods will be close to the defined significance level. Please refer to the **Figure~6 in the Appendix B.2.2** which contains the results (coverage rate, mean width of prediction interval and winkler score) for varying significance level (from 0.05 to 0.9) under Prophet model for AMZN dataset.
> > >
> > > "training different models or with different seeds and then testing the conformal method might have been a solution" - Regarding the error bars, our experimental settings strictly follow the settings of the prior methods (including ACI [1], PID [2], OGD [3], ECI [4]), where such metrics were not typically reported. We adhered to these settings to ensure full comparability with prior methods and to isolate the contribution of the proposed update rule from variations introduced by resampling. To address your concern, we conducted additional experiments on three synthetic datasets and evaluated each method under **five** different random seeds. We report both the mean and standard deviation of the coverage rate, interval width, and Winkler score. These results are included in the rebuttal above and in **Section 5.2.1** of the revised paper.
> > >
> > > Thank you for your constructive suggestions. If you have other questions, we are very happy to discuss them with you.
> > >
> > > [1] Gibbs, Isaac, and Emmanuel Candes. "Adaptive conformal inference under distribution shift." Advances in Neural Information Processing Systems 34 (2021): 1660-1672.
> > >
> > > [2] Angelopoulos, Anastasios, Emmanuel Candes, and Ryan J. Tibshirani. "Conformal pid control for time series prediction." Advances in neural information processing systems 36 (2023): 23047-23074.
> > >
> > > [3] Angelopoulos, Anastasios N., Rina Foygel Barber, and Stephen Bates. "Online conformal prediction with decaying step sizes." Proceedings of the 41st International Conference on Machine Learning. 2024.
> > >
> > > [4] Wu, Junxi, et al. "Error-quantified Conformal Inference for Time Series." The Thirteenth International Conference on Learning Representations.

---

### Author Response · Authors · 2025-12-01
**Responses to AC**

Dear AC, Thank you for your time and efforts to review our paper. According to reviewers' comments, we conducted additional experiments to address their questions and included these results in the revised version. Our responses to reviewers' comments are summarized below:

>Novelty

(1) We propose a dual-feedback mechanism that improves predictive efficiency and stabilizes online updates. It overcomes a key limitation of existing methods, which depend solely on the signal derived from the current threshold and consequently suffer from unstable and oscillatory update dynamics. This is the first framework to explicitly account for the influence of the conformal quantile threshold in online conformal area;

(2) We designed a residual-aware feedback function to provide fine-grained interval adjustments precisely, which is model-agnostic. Our differentiated design of the two feedback signals ensures that the update process remains smoother and more stable.

>Design of the estimated feedback function

The estimated feedback term is intended to stabilize and counterbalance the intervals, and its specific form is motivated by two theoretical considerations:

(1) For small deviations $|e_t^\*|$ (i.e., when the score is close to the estimated conformal threshold), the function $f(e_t^\*) = (1 - \frac{|e_t^\*|}{2B})\tanh(c e_t^\*)$ satisfies $\tanh(c e_t^\*) \approx c e_t^\*$, and therefore $f(e_t^\*) \approx c e_t^\*$ when it is small. This means that the estimated feedback behaves as a linear restoring force around the target quantile, producing a contracting update that corrects small mis-calibrations smoothly and avoids the oscillatory behaviour observed in existing methods.

(2) For large $|e_t^\*|$, we have $\tanh(c e_t^\*) \approx sign(e_t^\*)$, so that $f(e_t^\*) \approx (1 - \frac{|e_t^\*|}{2B}) sign(e_t^\*),$ which gradually attenuates to zero as $|e_t^\*| \to 2B$. This produces a redescending behaviour: extreme deviations are down-weighted so that the estimated feedback does not cause unnecessary expansions of the prediction interval, thereby improving efficiency without compromising stability.

>Experiments

In response to the reviewers’ comments, particularly those from Reviewer 68ke, we conducted additional experiments to address their questions and concerns.

(1) Section 5.2.1: We generated three synthetic datasets exhibiting different forms of non-stationarity. The results are averaged over 5 runs;

(2) Appendix B.2.2: We explored the performance of our method under varying significance levels. Meanwhile, we explored another feedback function softsign() as another squashing function;

(3) For all experiments, we used a new evaluation matrix (Winkler Score) and the results have been synchronized to all tables in the revised paper.

Overall, the experimental results demonstrate that DDCI achieves superior performance in most cases compared with all baseline online conformal methods.

---

### Meta-Review · Area_Chair_Euh3 · 2026-01-05

**Summary:**

Taking inspiration from control theory, the authors build on existing approaches to time-series conformal prediction, using estimated feedback in addition to actual feedback to obtain tighter intervals at given coverage levels. The proposed method is comprehensively evaluated across numerous benchmarks and generally achieves tighter intervals than baseline methods. While the method is well-motivated and evaluated, this paper would benefit from more in-depth analysis. In particular, the authors do not build significant intuition - either theoretically or through toy experiments - around why the estimated feedback provides additional utility that improves the conformal intervals. In the rebuttal, the authors tested some different toy settings of non-stationarity; however, these results failed to demonstrate the specific scenarios in which this additional feedback is most beneficial or potentially detrimental. This paper would benefit from a revision that spends more time analyzing and building intuitions around this additional signal, rather than just demonstrating better numbers across benchmarks.

**Reviewer Concerns:**

Unfortunately, only one of the reviewers responded to the authors' concerns. I did my best to account for this lack of communication while assessing the paper. While I believe the authors addressed almost all the detailed technical concerns, questions, and varying significance levels, the major issues around the lack of analysis (highlighted above) did not, in my opinion, feel sufficiently addressed.

**Reviewer Scores:**

- 68ke responded to the rebuttal with an additional question, though most of their concerns were addressed. It is possible that they would have increased their score, but the unenthusiastic tone in the response suggests to me that they would not have been convinced to vote for acceptance.
- The other reviewers did not engage at all during the discussion period, and so would likely not have changed their scores if they had had the full discussion period.

If the reviewers had engaged more, they might have found the author’s revisions satisfactory. I do commend the authors on a thorough response to the reviewers and a minor revision of their paper that addressed most concerns. However, based on my examination of the paper, I believe that, without assurance from the reviewers, it would be best for the authors to resubmit it after a major revision. I understand that this OpenReview debacle may have unfairly hurt the authors. Nevertheless, I do believe this paper has potential, and after another round of revision and submission, it will be a stronger paper that is more likely to have a strong impact on the community.

---

### Decision · Program_Chairs · 2026-01-26

Reject